# UV-curable thiol-ene system for broadband infrared transparent objects

Piaoran Ye [1,4], Zhihan Hong[1,4], Douglas A. Loy[2,3] & Rongguang Liang [1]✉

Conventional infrared transparent materials, including inorganic ceramic, glass, and sulfur-rich organic materials, are usually processed through thermal or mechanical progress. Here, we report a photo-curable liquid material based on a specially designed thiol-ene strategy, where the multithiols and divinyl oligomers were designed to contain only C, H, and S atoms. This approach ensures transparency in a wide range spectrum from visible light to mid-wave infrared (MWIR), and to long-wave infrared (LWIR). The refractive index, thermal properties, and mechanical properties of samples prepared by this thiol-ene resin were characterized. Objects transparent to LWIR and MWIR were fabricated by molding and two-photon 3D printing techniques. We demonstrated the potential of our material in a range of applications, including the fabrication of IR optics with high imaging resolution and the construction of micro-reactors for temperature monitoring. This UV-curable thiol-ene system provides a fast and convenient alternative for the fabrication of thin IR transparent objects.

Optics in the infrared (IR) region play a critical role in various applications, including cancer diagnosis[1], autopilot sensors[2,3], defense[4], and aerospace[5], etc. The most common materials used for IR windows and optics include ZnS, ZnSe, $GeO_2$, Ge, and Si[6], but most of these inorganic materials are brittle, have high melting points, and are costly to fabricate compared to most of polymeric materials[7,8]. Furthermore, many chalcogenide glasses are not transparent to visible light, which limits their applications[9]. In contrast, inorganic salts like NaCl, KBr, and $CaF_2$ can be used to fabricate IR windows with a relatively low-cost fabrication method such as high-pressure compression. However, they are sensitive to moisture in the air, which limits their usage environment.

Compared to most inorganic materials, organic polymeric materials are known for their good flexibility, extensibility, and plasticity at relatively low temperatures, which make processing easier compared to inorganic materials. Polymeric materials like poly(methyl methacrylate), epoxy resins, polycarbonate, and silicone have been widely used in different applications such as lens material, lens coating, and bandpass filters[10–12]. However, the majority of these polymers cannot be used for applications that require transparency in the MWIR (MWIR,

3–5 µm) or LWIR (LWIR, 7–14 µm) region due to the complex absorption peaks caused by covalent bonds in the polymer structure, resulting in a lack of transparency, especially when the polymer structure contains not only hydrocarbons but also oxygen, nitrogen, or other atoms that introduce dipole moments. This effect is more pronounced in the LWIR region since it is located in the fingerprint region of the IR spectrum. Poly(ethylene) (PE) shows good transparency not only in MWIR but also in LWIR due to its simple structure with only hydrocarbons[13]. However, because of its thermoplastic properties, this material has low thermal resistance. Furthermore, its semi-crystalline nature causes it to lose transparency in the visible light region.

To overcome the limitations of PE, some cross-linked polymers prepared using sulfur and vinyl compounds were developed. Elemental sulfur has a high refractive index and good transparency in both the MWIR and LWIR region, making incorporating a high ratio of sulfur into polymer chains a simple but effective route to fabricate IR-transparent materials[14]. Early reports describe utilizing inverse vulcanization with sulfur and diisopropenylbenzene (DIB) to form cross-linked copolymers that are transparent in the MWIR region[15]. However,

[1]Wyant College of Optical Sciences, The University of Arizona, 1630 E. University Blvd, Tucson, AZ 85721, USA. [2]Department of Chemistry & Biochemistry, The University of Arizona, 1306 E. University Blvd, Tucson, AZ 85721-0041, USA. [3]Department of Materials Science & Engineering, The University of Arizona, 1235 E. James E. Rogers Way, Tucson, AZ 85721-0012, USA. [4]These authors contributed equally: Piaoran Ye, Zhihan Hong. ✉e-mail: rliang@optics.arizona.edu

the presence of both methyl and aromatic rings in the network results in poor transparency of the material in the LWIR region. Efforts to enhance LWIR transparency, even in the presence of aromatic rings, have led to significant strides in recent research. Miyeon Lee et al. demonstrated the remarkable potential of using symmetric benzene-trithiol in conjunction with elemental sulfur, achieving peak transparency exceeding 70% in 1 mm thick samples[16]. Darryl Boyd et al. explored the influence of comonomer isomers of divinylbenzene (DVB) on the optical properties of inverse vulcanization products, revealing a promising avenue. By opting for DVB isomers over DIB, bulk polymers (>1 mm thickness) showcased noteworthy LWIR transmission capabilities, approaching approximately 10% transparency[17] Conversely, Tristan S. Kleine et al. reported that replacing the styrene with oligomers prepared using norbornadiene improved the transparency in the LWIR region since no more asymmetrical aromatic rings exist in the final polymer[18-20]. Norbornadiene and organo tin molecules that have four vinyl groups were also used with sulfur to obtain the polymer transparent in the LWIR region[21]. One limitation of inverse vulcanization is that it requires an elevated temperature (usually between 150 and 200 °C) to open the cyclic sulfur ring and finish the free radical polymerization, which means it can only be cured by thermal processes[22].

Compared to thermal curing, photo-curing shows advantages in curing efficiency and convenience. Moreover, the rapid development of photo-based 3D printing techniques further enhances the potential of photo-curable systems in the fabrication of objects with tiny and complex structures. Optics with complex structures have been fabricated through different types of photo-based 3D printing techniques[23]. However, no material has been reported that is transparent in both MWIR and LWIR regions and can be 3D printed through a photo-curing mechanism. Therefore, it would be desirable to develop a photo-curable resin that is transparent in multiple IR regions.

Thiol-ene click reaction is famous for its high efficiency[24]. This reaction mechanism, operating through a free radical process, offers versatility by being triggered through either thermal or photo initiation, both with or without initiators[25-29] Such feature not only makes the thiol-ene system as a powerful tool in polymer synthesis and biosynthesis[30], but also makes it widely used in the material science including coating[31], molding[32], and additive manufacturing[33-35]. Notably, the majority of thiol-ene materials exhibit favorable transparency within the visible light range, rendering them a prime choice for fabricating optics characterized by diverse mechanical and optical properties[36-39].

However, the most thiol-ene systems are unsuitable for infrared imaging applications, especially within the LWIR region. This is primarily due to the presence of molecular structures, such as ether and carbonyl groups, which lead to IR absorption across both MWIR and LWIR spectra, resulting in diminished transparency[19]. A noteworthy endeavor by Yang Qiu et al. involves imprinting a high refractive index Fresnel Lens structure onto silicon or quartz surfaces using a trithiol monomer[40]. This system's absence of oxygen allows it to exhibit transparency in both the MWIR (200 μm thickness, 62% peak transparency) and LWIR (200 μm thickness, 24% peak transparency) regions. This work utilized a low molecular weight trithiol with allyl, styryl and propargyl modified thiols. Additionally, the utilization of photo-based additive manufacturing to fabricate optics transparent in both MWIR and LWIR regions has yet to be explored.

In this work, we report a thiol-ene system that can be efficiently cured by UV or deep blue light. The cured materials possess good elasticity and demonstrate workable transparency in both the MWIR and LWIR regions, while also maintaining transparency in visible light. Supplementary Table 1 offers a comparison between our reported systems and some reported organic IR-transparent materials. IR transparent optics were fabricated at room temperature within minutes with a simple molding method assisted with UV irradiation.

Moreover, microlens array works in the IR region were 3D printed using a customed laser direct writing system equipped with a 780 nm femtosecond laser[41]. The imaging performance of fabricated optics shows good sensitivity and imaging resolution in different IR regions. In addition, we also used this type of thiol-ene resin to print a microreactor which can be used to monitor the reaction temperature with the assistance of an LWIR camera due to the good LWIR transparency of the thin wall of the reactor.

## Results

### Molecule choice and design
As previously mentioned, PE has excellent IR transparency in both the LWIR and MWIR ranges, as depicted in Supplementary Fig. 9. This is due to the PE molecule consisting only of C and H atoms, which avoids the vibration caused by bonds between C (or H) and other hetero atoms, such as C = O, C-O, C-N, C-X, N-H, O-H, etc. Additionally, saturated hydrocarbons avoid vibrations caused by alkenes, alkynes, or aromatic rings. Moreover, the highly branched molecular structure of LDPE forces the methylene chains apart, making the $CH_2$ peaks of LDPE in the fingerprint region simpler than the $CH_2$ peaks of HDPE in the fingerprint region, resulting in better transparency in the LWIR than HDPE[42]. Some previously reported LWIR transparent materials contain not only C and H, but also S and Tin[18,21]. The absorption peaks caused by C-S and C-Sn are not as complicated as the peaks caused by C-O stretching in the fingerprint region, which helps avoid massive peaks in the LWIR region. Therefore, to achieve IR transparency in both the MWIR and LWIR range, the molecular design follows two basic principles: 1. The element (atom) types in the final molecular structure should be kept to a minimum; 2. The asymmetric aromatic rings should be avoided in the final molecular structure. As UV-curing is a desired feature of the target material, the thiol-ene curing system was selected because it allows avoiding the use of acrylate, methacrylate, or vinyl benzene, which are also UV-curable but introduce chemical bonds that significantly affect the MWIR and LWIR transparency.

To enable the thiol-ene system for 3D printing, multifunctional building blocks with more than two functional groups per molecule are required to form the crosslinked network during the printing process. However, commonly used candidates in thiol-ene curing systems, such as glyoxal bis(diallyl acetal), pentaerythritol tetrakis(3-mercaptopropionate), Trimethylolpropane tris(3-mercaptopropionate), and 1,3,5-Triallyl-1,3,5-triazine-2,4,6(1H,3H,5H)-trione were not selected for preparing the IR transparent resin, as they all contain C-O, C = O, or C-N bonds. Instead, tetrathiol and polythiol, which only contain C, H, and S, were chosen as the source of thiol. The di-vinyl oligomers (DVO) were synthesized using two equivalents of 5-vinyl-2-norbornene and one equivalent of dithiols. Due to the reactivity difference between the vinyl in the norbornene ring and the terminal vinyl outside of the ring, the thiol group preferred to react with the vinyl in the norbornene ring, leaving the terminal vinyl unreacted[43]. The tetrathiol (tetraSH) and polythiol (polySH) were then synthesized and utilized as the multi-thiol source in the thiol-ene reaction during UV curing. Figure 1a shows the basic route designed to prepare the thiol-ene system for the IR transparent resin.

To prepare the DVO required for the thiol-ene reaction, four different dithiols, namely 2,2'-thiodiethanol, 1,6-hexanedithiol, 1,8-octanedithiol, and 1,10-decanedithiol, were chosen as bridge molecules to react with 5-vinyl-2-norbornene, yielding DVO2, DVO6, DVO8, and DVO10, respectively (Fig. 1b). It is interesting to note that the crosslinked material prepared with DVO containing longer hydrocarbon bridges showed lower transmission percentages in the LWIR region. Specifically, samples prepared with DVO6, DVO8, and DVO10, when measured with a 500 μm thickness window, exhibited very low transparency with no peak higher than 5% of transmission in the LWIR region (Fig. 1d).

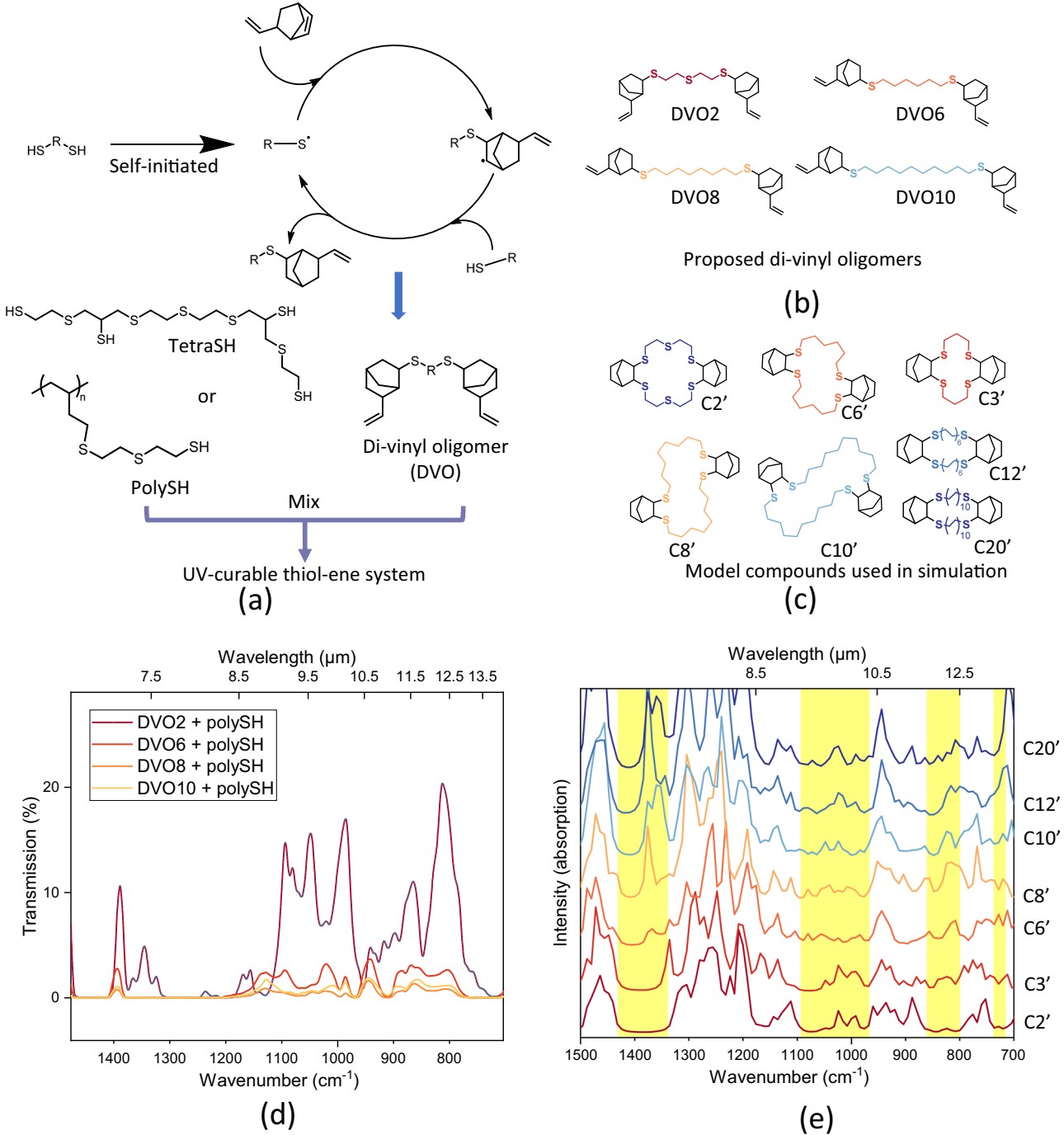

**Fig. 1 | Molecular structures of multithiol-divinyl oligomer (DVO) system and the long-wave infrared (LWIR) transmission results for pre-screening. a** Scheme of monomer and resin preparation for the study; (**b**) proposed di-vinyl oligomers(DVO) that will be selected; (**c**) model compounds that used in computation.

**d** Experimental LWIR transmission percentage of samples (500 μm thickness) prepared using DVO(2-10) and polySH; (**e**) simulated LWIR intensity of compounds in (**c**). The yellow regions refer to potential windows transparent to LWIR. Source data are provided as a Source Data file.

Since the samples used in the study are not purified, to better understand and verify the transparency trend along with the carbon numbers, computations of some model compounds were also conducted to predict the IR spectra. Computational chemistry can be a powerful tool to predict the transmission performance of a material in the LWIR region[18,20]. In this study, several model compounds were designed based on the structure of different dithiols, and the simulated IR spectra of gas phase were generated through a standard computational procedure (see supporting information). The different dithiol molecules were connected through two norbornane rings to simplify the modeling molecules. Spectra of compounds that have 3,

12, and 20 carbons between S atoms were also simulated. Since most of the MWIR region refers to wavenumber higher than 1500 cm⁻¹, which is out of fingerprint region, the absorption peaks within the MWIR region can be easily predicted from the molecule structure, and all the proposed candidates should have similar absorption peaks in MWIR region. Therefore, only the computational results of the LWIR region were compared. Figures 1b, c shows the correspondence between the potential molecules designed for IR transparent resin and the model compounds used in the simulation.

It is important to note that the gas phase IR spectra predicted by computational simulations do not necessarily reflect the real

transmission spectra of condensed phase samples, particularly when the samples are relatively thick. Gas phase spectra represent extremely dilute samples, while in the condensed phase, intermolecular interactions between molecules can lead to more complex absorption peaks that cannot be ignored[13]. Therefore, the main purpose of simulating gas phase spectra is to identify a window containing zero or weak IR signals. The fewer and weaker the peaks are, the greater the likelihood of achieving high transmission for the corresponding wavelength when the sample is in a condensed phase with a relatively thick thickness.

The simulated IR spectra showed that when there were only two carbons between sulfur atoms, four small windows were identified that did not contain significant absorption peaks (yellow region in Fig. 1e). These windows aligned with the high transmission region (Fig. 1c) observed in the actual sample in Fig. 1d. However, as the number of carbons increased to three, new absorption peaks appeared in the 800 cm$^{-1}$–860 cm$^{-1}$ window and the 720 cm$^{-1}$ window, and the 1360 cm$^{-1}$–1425 cm$^{-1}$ window became narrower. The introduction of new hydrocarbon bonds, which differ from those directly connected to the S atoms, caused this effect. For example, the new peak at around 1350 cm$^{-1}$ could belong to out-of-plane -CH$_2$- rocking vibrations when there are more than 3 carbons between two sulfur atoms. Further increases in the number of carbons to 6 and 8 introduced more complex C-H vibrations in the fingerprint region, causing the 1360 cm$^{-1}$–1425 cm$^{-1}$ window to decrease in size and the 1050 cm$^{-1}$–1080 cm$^{-1}$ and 800 cm$^{-1}$–860 cm$^{-1}$ windows to disappear. Increasing the number of carbons to 10, 12, and 20 did not lower the signal intensity in the LWIR region. While a longer backchain similar to HDPE may improve LWIR transparency, the use of a dithiol with an excessively long carbon chain could lower the thiol-ene reaction rate since the long carbon chain diluted the concentration of thiol and vinyl functional groups in the resin, which is not ideal for a UV curing system, particularly for 3D printing. Therefore, 2,2′-thiodiethanol (TDE) was chosen for the synthesis of both DVO and the multi-thiols for preparing IR transparent films and optics.

The synthesis of DVO was achieved by mixing TDE and 5-Vinyl-2-norbornene, which then reacted with each other. The thiol group exhibited a preference for reacting with the norbornene ring due to the difference in reactivity between the vinyl in the ring and the terminal vinyl outside the ring. As a result, the terminal vinyls were left unreacted, although a small portion of the thiol group still reacted with them. The 1H NMR spectrum indicated that during the synthesis of DVO2, approximately 75% of ring vinyls (peaks from 6.15 ppm to 5.90 ppm) and 25% of terminal vinyls (peaks from 5.80 ppm to 4.75 ppm) had reacted with thiols (Supplementary Fig. 8). However, this is unlikely to have any impact on the final curing process since the remaining ring vinyls can react with thiols during UV curing.

To prepare the resin for two-photon printing and UV curing, the vinyl and thiol concentrations of DVO, polySH, and tetraSH were determined by NMR or titration (see supporting information for details), as they were not highly pure reagents. Then, DVO2 was mixed with either polySH or tetraSH. Inhibitor and photo-initiator were added, with 0.1 wt% of pyrogallol and 2 wt% of 2,4-diethyl-9H-thioxanthen-9-one used as the respective additives. Conventional two-photon initiators, such as 4,4′-bis(dimethylamino)benzophenone and 4,4′-bis(diethylamino)benzophenone, were not suitable for the described thiol-ene system as they caused severe retardation during curing due to deprotonation of the thiols by amine groups in the initiators[44]. On the other hand, 2,4-diethyl-9H-thioxanthen-9-one was found to be a suitable photo-initiator as it did not contain amine groups and had acceptable initiation efficiency despite its lower absorption efficiency to 380 nm light compared to other initiators.

## IR transparency of samples in LWIR and MWIR range

To assess the infrared (IR) transparency of samples prepared with tetraSH or polySH, two sets of test windows with thicknesses of approximately 150 µm and 550 µm were fabricated on KBr crystals. The 150 µm samples demonstrated good transparency LWIR region, as two large windows (7–8 µm and 8.5–14 µm) with transmittance as high as 70% were observed (Figs. 2a, b). However, with a thickness of 500 µm, the transmittance decreased. Both samples exhibited three main transparent windows in the LWIR region, namely, 1313 cm$^{-1}$–1401 cm$^{-1}$, 1124 cm$^{-1}$–960 cm$^{-1}$, and 954 cm$^{-1}$–760 cm$^{-1}$, with transmittance between 10% and 20%. At a thickness of around 900 µm, both materials demonstrated only minor transmission peaks in the LWIR region, showing nearly no transparency to LWIR at this thickness. For comparison, two additional windows were produced using diurethane dimethacrylate (DUDMA, Fig. 2c) and pentaerythritol tetraacrylate (PETA, Fig. 2d), which are commonly utilized as monomers or crosslinkers in 3D printable resins. Due to strong absorption by C-O, the transparency of windows fabricated using these two molecules in the LWIR region was considerably poorer (Figs. 2c, d). With a thickness of 150 µm, DUDMA demonstrated only a narrow window from 10.5 to 14 µm with a peak transmittance of 30%, while PETA exhibited better performance, with a peak transmittance of around 55%, but the transparency window was still much narrower than that of samples produced using tetraSH and polySH in conjunction with DVO2. At higher thicknesses (~500 µm), both cured DUDMA and PETA demonstrated almost no transparency in the LWIR region. The reported transparency results suggest the potential of our resin in LWIR applications requiring compact optical systems[45,46].

In addition to the LWIR transparency of the tetraSH and polySH samples, it is also worth noting that they exhibit decent transparency in the MWIR region (3–5 µm). To evaluate the MWIR transparency, samples prepared with tetraSH or polySH together with DVO2, DUDMA, and PETA were compared. Since all those materials have much fewer absorption peaks between 3–6.5 µm compared to the absorption peaks in the LWIR region, they all show better transparency in the MWIR region with a thickness of 500 µm. However, DUDMA and PETA contain high amounts of C = O which severely affects their transparency in the MWIR region. Both the tetraSH-DVO2 and polySH-DVO2 samples show a maximum transmission of over 60% (Figs. 2e, f), while the DUDMA and PETA samples only show transmission of less than 30% and 20%, respectively (Figs. 2g, h). Even at a thickness of 2 mm, the tetraSH and polySH samples exhibit a maximum transmission of close to 20%, whereas the DUDMA and PETA samples show almost no transmission. These results highlight the advantages of using our thiol-ene system for MWIR transparency compared to conventional UV-curable materials.

## Thermal and mechanical properties of cured resin

Prior to performing 3D printing or molding, the mechanical and thermal properties of the UV-cured resins were characterized. It was observed that both samples prepared using tetraSH or polySH exhibit chemical stability at temperatures below 300 °C (Supplementary Fig. 10). Upon UV curing, the resins displayed significant elastic behavior under external pressure, which indicates that they are likely elastomers with glass transition temperatures ($T_g$) lower than room temperature. The Tg of the two resin systems was determined using DSC (Fig. 3a) and DMA (Fig. 3b). The results from both methods indicated that the resin prepared using tetraSH or polySH have slightly different $T_g$, with a slightly higher $T_g$ observed for the polySH-prepared sample. Meanwhile, the discrepancy between the DSC and DMA results could be due to the difference in thermal transportation in the two technologies, as well as the difference in sample size used in DMA and DSC. However, both sets of results indicate that the cured resin can be considered an elastomer at room or higher temperatures.

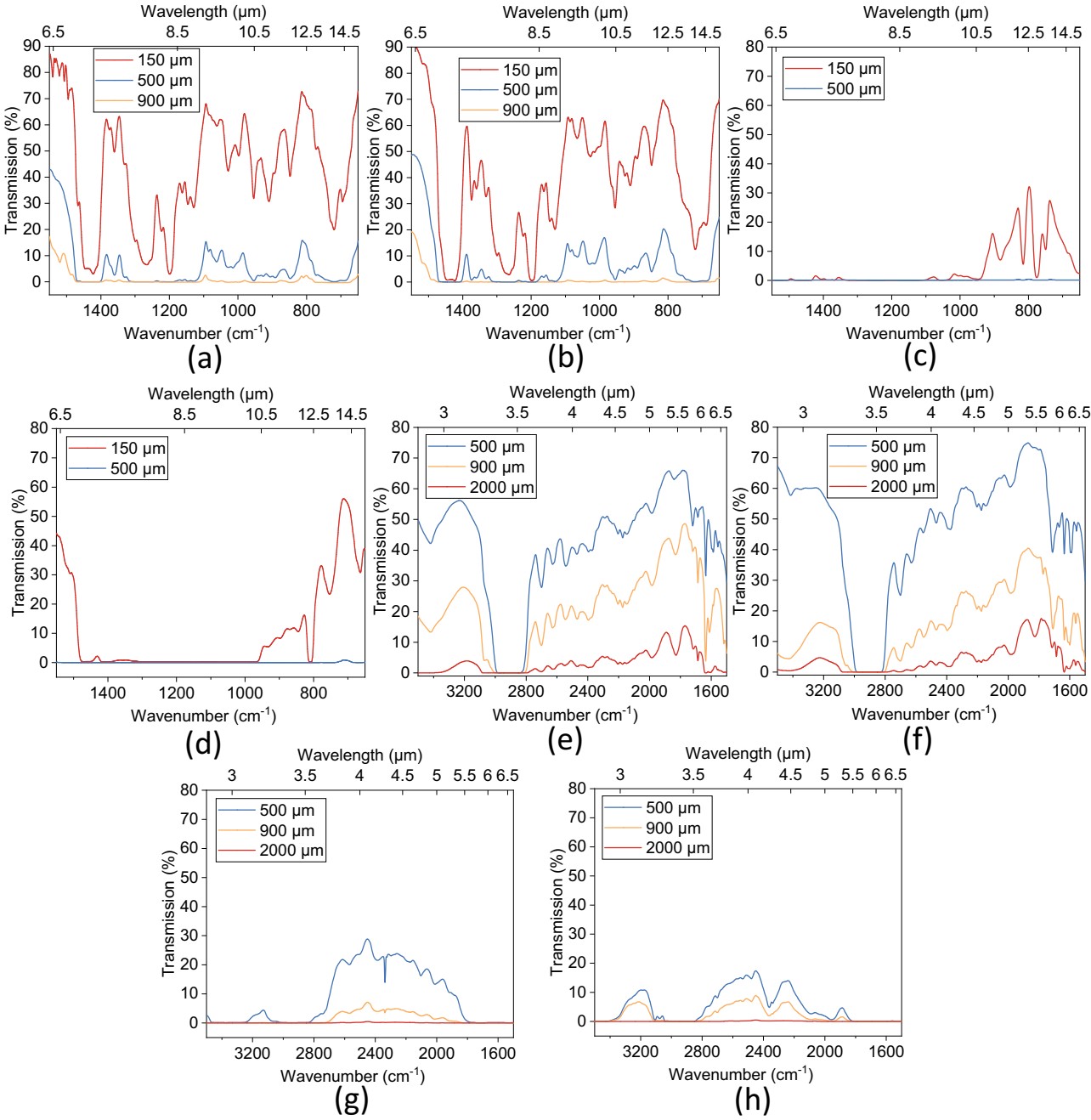

**Fig. 2 | Broad band IR transmission results of samples with different thickness.** Long-wave infrared (LWIR) transmission of samples prepared using (**a**) TetraSH with divinyl oligomer (DVO2), (**b**) polySH with DVO2, (**c**) diurethane dimethacrylate (DUDMA), and (**d**) pentaerythritol tetraacrylate (PETA) with different thicknesses. Mid-wave infrared (MWIR) transmission of samples prepared using (**e**) tetraSH with DVO2, (**f**) polySH with DVO2, (**g**) DUDMA, and (**h**) PETA. Source data are provided as a Source Data file.

It is important to note that during some early DMA testing, it was observed that the storage modulus increased at temperatures higher than 140 °C (Supplementary Fig. 11a), suggesting the possibility of additional crosslinking during the measurement. This phenomenon was more pronounced in samples prepared using polySH. Correspondingly, a broad exothermic peak was also observed in the DSC measurement of cured polySH-DVO2 samples at temperatures higher than 150 °C (Supplementary Fig. 12a), which disappeared upon the second run of the same sample. However, this peak was not significant in the DSC measurement of cured tetraSH-DVO2 samples (Supplementary Fig. 12b), indicating that tetraSH-DVO2 samples have higher monomer conversion during UV curing. FTIR of the UV-cured sample also revealed the presence of unreacted vinyl and thiol

(Supplementary Fig. 13), which is consistent with previous literature on thiol-ene systems that conversion may not reach 100% during UV-curing[47,48].

To enhance our comprehension of the curing kinetics and optimize thiol-ene conversion, we employed FTIR to monitor the consumption of both thiols and alkenes during UV exposure (Supplementary Fig. 13). Our findings indicate that in both the tetraSH and polySH systems, the conversion of thiol rapidly stabilizes at approximately 70% (thiol) for tetraSH and 50% (thiol) for polySH within the initial 60 seconds of standard UV exposure (Supplementary Figs. 13e, 13f). These conversions are maintained over the subsequent 9-min period. In order to investigate the potential for post-UV curing to further enhance thiol-ene conversion, we

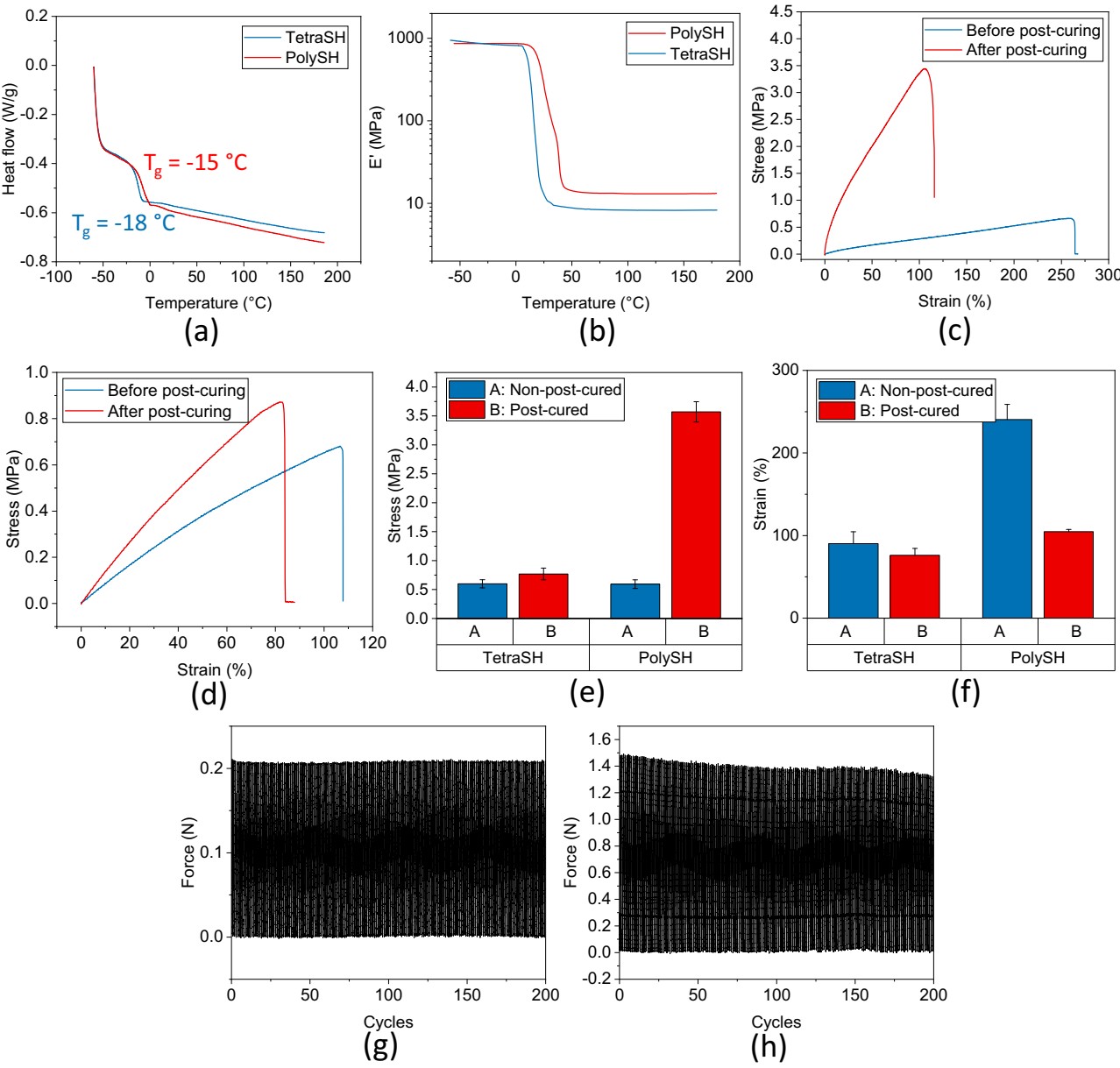

**Fig. 3 | Thermal and mechanical properties of cured tetraSH-divinyl oligomer (DVO2) and polySH-DVO2. a** Differential scanning calorimetry (DSC) and (**b**) dynamic mechanical analysis (DMA) of samples after thermally post-curing treatment; (**c**) stress-strain curve of samples prepared with tetraSH and DVO2; (**d**) stress-strain curve of samples prepared with polySH and DVO2; (**e**) comparison of samples' ultimate stress before and after thermally post-curing. Error bars: Standard Error; (**f**) comparison of samples' ultimate strain before and after thermally post-curing. Error bars: Standard Error; (**g**) cyclic tensile testing of samples prepared with tetraSH and DVO2; (**h**) cyclic tensile testing of samples prepared with polySH and DVO2. Source data are provided as a Source Data file.

conducted two additional post-UV curing steps. Initially, we extended the UV exposure by another 10 minutes while maintaining the same power. The tetraSH system exhibited consistent conversion rates for both thiol and alkene, whereas the polySH system demonstrated a slightly increased conversion (to ~55% for thiol). Subsequently, we subjected the samples to additional curing using a Formlabs post-curing machine (at room temperature for 12 h). The tetraSH system's conversion remained unchanged, whereas the polySH system's thiol-ene conversion significantly increased to approximately 90% (thiol), highlighting distinctive UV curing kinetics for these two systems. This difference can be attributed to structural dissimilarities between tetraSH and polySH, as the latter contains only primary thiols while the former comprises both primary and secondary thiols.

During the initial 10-minute curing period, the tetraSH shows a higher reaction rate in comparison to the polySH system which might be due to the lower molecular weight and viscosity of tetraSH compared to polySH. However, the lower reactivity of secondary thiols in tetraSH compared to primary thiols becomes a limiting factor[49]. As the curing process progresses, the constrained mobility impedes the remaining secondary thiols from reacting effectively with alkenes under UV conditions. Conversely, in the polySH system, unreacted primary thiols continue to react with alkenes, albeit at a relatively slower rate. This continuous reaction capability enables the conversion to reach 87% through overnight UV exposure. Nevertheless, while the DSC results for both the polySH-DVO2 and tetraSH-DVO2 systems display a nearly flat curve between 100 to 200 °C (Supplementary Fig. 12c), DMA reveals a slight elevation in E' values for both systems at

temperatures exceeding 160 °C (Supplementary Fig. 11b). This suggests a potential for further improvement in thiol-ene conversion.

Besides UV curing, it is well-known that thiol-ene system can be cured by thermal curing with and without initiators[26,27,50]. To further maximize thiol-ene conversion, both systems were subjected to a post-curing thermal treatment at 170 °C for 12 h. After post-curing, both DMA and DSC measurements showed stable properties of the samples at temperatures from 100 to 200 °C, indicating that the reachable thiols and vinyls had reacted (Figs. 3a, b). The storage modulus also increased from 3.8 MPa to 8.3 MPa for tetraSH-DVO2 samples and from 1.3 MPa to 13.2 MPa for polySH-DVO2 samples, respectively. Correspondingly, the thiol conversion for tetraSH-DVO2 and polySH-DVO2 were increased to 91% and 98%, respectively, as the thiol and alkene peaks became almost invisible in FTIR spectra (Supplementary Figs. 13a-d). Here, the thiol conversion of tetraSH-DVO2 also increased indicating that the increased temperature helps increase the molecule mobility which makes secondary thiol more easier to react with alkene. Although FTIR still showed unreacted thiol and vinyl peaks for both systems after post-curing (Supplementary Fig. 13, Supplementary Fig. 14), these vinyls and thiols are probably trapped and not reachable from each other since they are embedded in a solid. We observed that the thermal post-curing slightly increased the transmission in the LWIR region (Supplementary Figs. 14c, 4d), but this difference is not significant enough to improve the imaging quality in the later imaging test. It's also important to note that thermally post-curing in the air can cause oxidation of the sample, leading to brown color and decreased transparency in the visible light region (Supplementary Fig. 15)[51]. This color change was more noticeable for the sample prepared using polySH and DVO2. This oxidation can be avoided by post-curing the sample in a nitrogen atmosphere resulting in a sample without brown color.

In order to assess the mechanical properties of the resin after UV curing, coupons molded into ASTM D638 type IV were prepared using either tetraSH or polySH. These coupons were then subjected to stress-strain testing, both with and without thermal post-curing. The stress-strain curves exhibited elastomeric behavior. Although both tetraSH-DVO2 (Fig. 3c) and polySH-DVO2 (Fig. 3d) samples had similar ultimate stress values, the samples made using polySH showed a much higher ultimate strain of 240% compared to tetraSH's ultimate strain of around 90%. This was attributed to the longer backbone of the polybutadiene used for the synthesis of polySH, which resulted in a more flexible final network. Additionally, the stress-strain results showed that the samples prepared with polySH were tougher than those prepared with tetraSH. Subsequently, both sets of samples underwent post-curing, resulting in the formation of more rigid networks. The ultimate stress was increased (from 0.6 MPa to 0.77 MPa for tetraSH-DVO2 and from 0.6 MPa to 3.57 MPa for polySH-DVO2), while the ultimate strain was decreased (from 90% to 76% for tetraSH-DVO2 and from 240% to 105% for polySH-DVO2). These results were consistent with DMA and DSC findings and confirmed that more thiols and vinyls reacted during the post-curing process. Table 1 lists major thermal and mechanical properties of both tetraSH-DVO2 and polySH-DVO2 systems. Furthermore, cyclic tensile testing was performed on both tetraSH-DVO2 (Fig. 3g) and polySH-DVO2 (Fig. 3f) samples. TetraSH-DVO2 samples show good recovery ability after 200 cycles when the maximum strain was below 30%. However, larger strains, such as 50%, caused the samples to weaken and ultimately break, likely due to tiny

defects introduced during the 3D printing of the ASTM D638 type IV molds or during coupon molding. Meanwhile, PolySH-DVO2 samples show a slightly decreasing maximum force as the cycles increased with some small damages observed. This could be due to the fact that the overall monomer conversion of polySH-DVO2 is lower than the monomer conversion of tetraSH-DVO2 making it easier to be damaged under external stress.

## Imaging performance in the MWIR range

We used both molding and two-photon printing methods to fabricate optics with low surface roughness (<5 nm) and high surface accuracy (average error <1 μm) (see methods section for more fabrication details). To evaluate the MWIR imaging performance of both molded and printed optics, we utilized an MWIR camera. The original IR lens of the camera was replaced with the molded lens, which was mounted with the assistance of a stand. To conduct the imaging experiment with a transmission strategy (Fig. 4a top), we used a laser-cut PMMA as an IR mask with a thickness of approximately 1 mm that blocked most of the MWIR signal from the 40 °C blackbody. The imaging results for the lens molded by either tetraSH-DVO2 or polySH-DVO2 are shown in Figs. 4c, d. These results demonstrated the ability of the lens to capture MWIR images with sharp edges, as evidenced by the capture of the A logo and the tiny dot structure on the outer contour caused by laser cutting. Although both tetraSH-DVO2 and polySH-DVO2 lenses exhibited lower contrast compared to the commercial lens (Fig. 4b), they exhibited much better imaging quality compared to the lens made with PETA (Fig. 4e), which absorbed much more MWIR and led to a blurry image result. Additionally, Fig. 4g demonstrated the competitive imaging resolution of the molded lens compared to the commercial IR lens (Fig. 4h), albeit with slightly lower contrast. These results suggest that our material has great potential to fabricate compact imaging systems, as all these images were taken with molded lenses of only 6 mm diameter and 180 μm thickness. Furthermore, our material's transparency to visible light provides an additional advantage over conventional IR lens materials, allowing for more compact designs in applications that require detecting both visible and IR light[52,53].

To further demonstrate the imaging resolution of the molded IR lenses, we utilized the USAF 1951 target to reflect the thermal radiation from an 80 °C heat plate (Fig. 4a bottom). The USAF target body is made of glass that absorbs MWIR, while all the numbers and marks on this target are coated with chrome, which reflects MWIR with high efficiency. Therefore, all the numbers and marks show a much brighter color than other areas of this target in an MWIR image. Figure 4h clearly shows that both the 3rd vertical and horizontal elements under group 3 can be identified, which have a line width of 49.61 μm, indicating excellent imaging resolution considering the wavelength of MWIR applications. It should be noted that the scattering from the reflection may lower the contrast, and the tilted direction (parallel) may decrease the image resolution in the perpendicular axis.

The performance of the 3D-printed lens array was also evaluated. To prevent light from passing through the gaps between each singlet lens in the lens array, a plastic mask was also 3D printed using commercial printing resin to cover the lens array. This mask absorbs MWIR and improves the imaging quality (Fig. 4i). Furthermore, a black mask was also used to cover the entire NaCl substrate except the lens array region to ensure that the IR light only passes through each singlet lens

**Table 1 | Summary of thermal and mechanical properties of reported thiol-ene systems**

| | $T_{d10}$ | $T_g$ | | Stress at break (MPa) | | Strain at break (%) | | E' (MPa) |
|---|---|---|---|---|---|---|---|---|
| | | DSC | DMA | Before thermal curing | After thermal curing | Before thermal curing | After thermal curing | |
| tetraSH-DVO2 | 291 °C | ~-18 °C | ~-7 °C | 0.60 ± 0.07 | 0.77 ± 0.10 | 90.17 ± 14.43 | 76.05 ± 8.30 | 8.3 |
| PolySH-DVO2 | 304 °C | ~-15 °C | ~-12 °C | 0.59 ± 0.07 | 3.57 ± 0.17 | 240.38 ± 18.39 | 104.75 ± 2.70 | 13.2 |

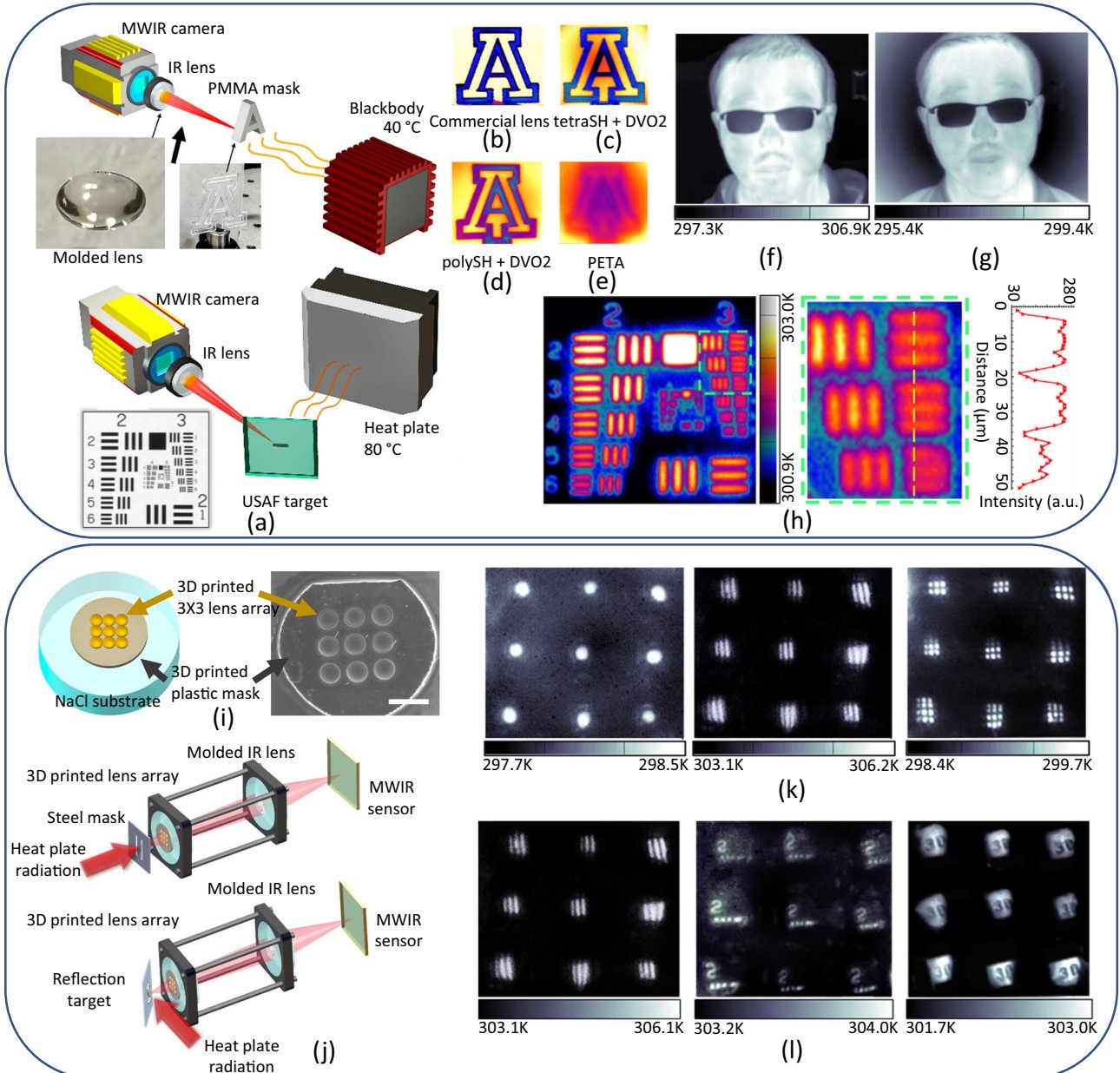

**Fig. 4 | Mid-wave infrared (MWIR) imaging performance of lenses fabricated by molding or 3D printing.** Sc heme of set up for MWIR imaging experiments with (**a**) transmission strategy and set up for imaging experiments of USAF 1951 target with reflection strategy. **b**–**e** Imaging taken with commercial lens, lens molded with tetraSH and divinyl oligomer (DVO2) resin, lens molded with polySH and DVO2 resin, and lens molded with PETA, respectively. All the molded lenses have a diameter of 6 mm and a thickness of 180 μm. **f** Image of human face obtained with MWIR sensor and a commercial lens. **g** Image of human face obtained with MWIR sensor and molded lens. **h** The reflection imaging result of a USAF 1951 target as a reflective target. The main body of the USAF target is made of glass and the numbers and marks on this target are coated with chrome with high reflectivity of IR light. The middle image in the dashed green box is a magnification of the region in the dashed green box in the left image. The inset contrast intensity profile (right panel) refers to three elements covered by the yellow dashed line on the middle

image. **i** Scheme of fabrication of lens array for the experiment. A 3 × 3 lens array was printed on a NaCl plate by two-photon polymerization using resin prepared with tetraSH and DVO2. A polymer mask containing 3X3 holes fitting to the lens array was 3D printed by a commercial digital light processing (DLP) printer and commercial resin to block light that does not pass through the lens array. Scale bar: 1 mm. **j** Scheme of the imaging system for experiments. The printed lens array and molded lens were assembled in a frame to maximize the imaging quality. The setup with black tape mask was shown in Supplementary Fig. 17. Both the transmission strategy (top) and reflection strategy (bottom) were employed during the experiments. **k** Imaging results obtained using transmission strategy of steel masks with single hole (left, 5 mm diameter), vertical grille (middle, 2 mm width), and mesh grille (right, 2 ×2 mm for every single lattice). **l** Imaging results obtained using reflection strategy of USAF target (left and middle) and a steel ruler (right) as reflection targets.

in the lens array. To further enhance the imaging quality, the printed lens array was assembled with the molded lens (tetraSH-DVO2) to create an optical system (Fig. 4j). Importantly, all the imaging optics in this system were made from our material, rather than commercial IR optics.

To demonstrate the imaging capabilities of the assembled imaging system, both transmission and reflection strategies were employed (Fig. 4j). Various steel masks with different shapes were utilized to evaluate the imaging quality of the system (Fig. 4k). The results demonstrated that objects with millimeter-sized dimensions

could be captured with clarity. However, some images exhibited distortion, which may have resulted from errors during the printing process of some singlet lenses.

To quantitatively assess the imaging resolution of the assembled imaging system, we employed the USAF target as a positive target to reflect the radiation from a heat plate (Fig. 4l). Our analysis revealed that element 2 (1.78 mm line width) under group -2 could be identified, highlighting the high-resolution capabilities of the system. In addition, we also used a steel ruler as a negative reflection target to conduct the same experiment, as the numbers on this ruler are a poor region for

reflecting the MWIR. Both 3 and 0 in Fig. 4l had widths of around 3 mm and could be observed through the assembled imaging system.

## Imaging performance in the LWIR range

To demonstrate the LWIR imaging ability of the optics fabricated using multithiol-DVO resin, a 3D printed mount was utilized to assemble the molded lens onto an LWIR sensor from Seek Thermal®, which is sensitive to 7.8–14 μm light. This mounting structure allowed for convenient lens changes within the imaging system (Fig. 5a). Comparison pictures taken by a molded tetraSH-DVO2 lens and a PETA lens with the

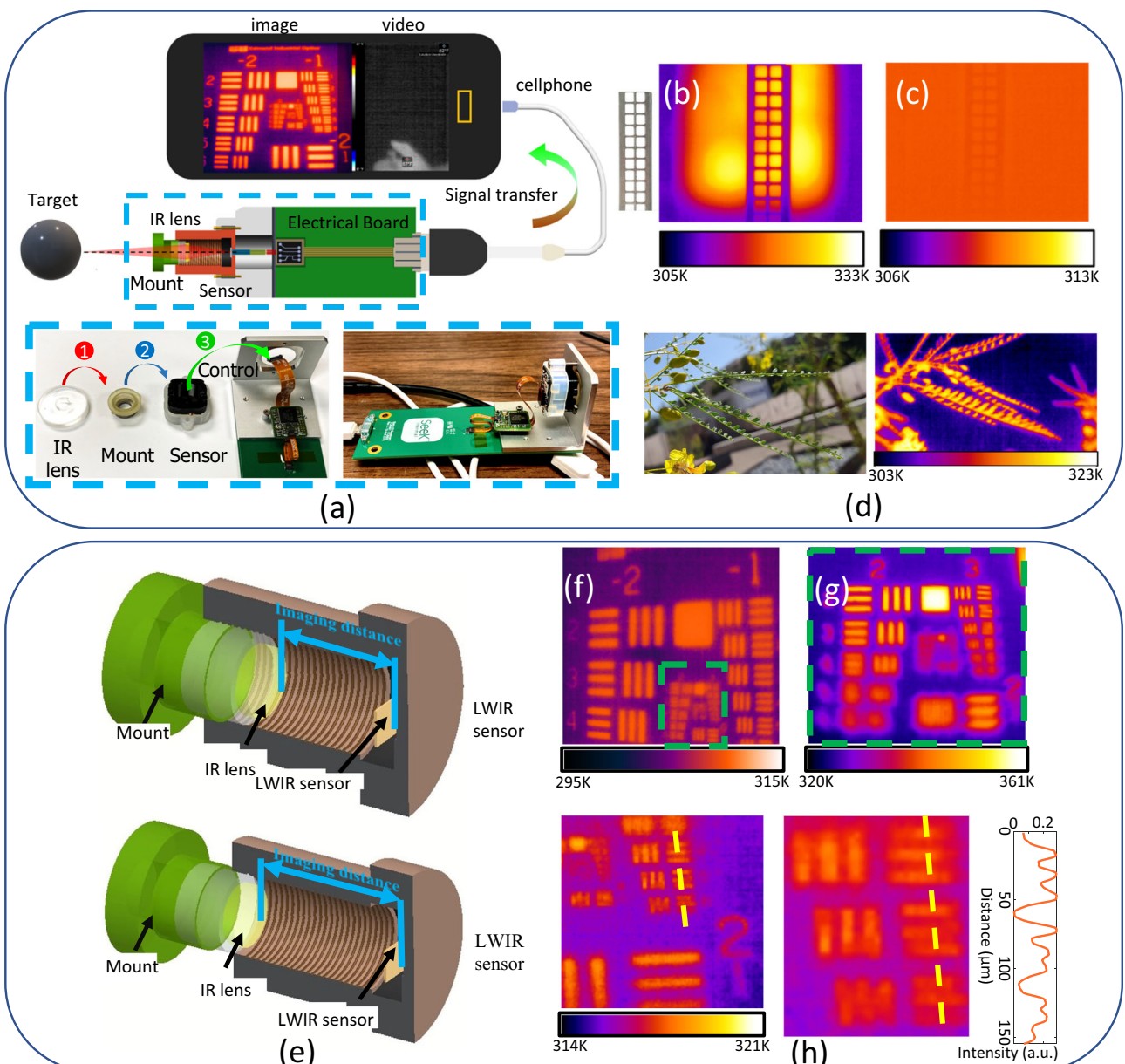

**Fig. 5 | Long-wave infrared (LWIR) imaging performance of lenses fabricated by molding. a** Scheme of assembling LWIR camera using the molded lens. The assembled camera can connect to a cellphone to capture images or record video using an APP provided by Seek Thermal. **b** LWIR imaging result of a mesh grille using lens molded using tetraSH and divinyl oligomer (DVO2). **c** LWIR imaging result of a mesh grille using lens molded using pentaerythritol tetraacrylate (PETA). **d** LWIR imaging result of a tree under sun exposure using lens molded by polySH and DVO2. The left picture is a visible light picture taken by a cellphone. The slight difference between visible and IR pictures was due to the small difference in field of

view and the photographing angles between the cellphone and the assembled IR camera. **e** Scheme of adjusting imaging distance for LWIR imaging using the 3D printed mounting structure. **f–h** The field of view (FOV) and magnification change as the imaging distance increases. The imaging experiments were conducted with the reflection strategy. Panel (**g**) refers to the region in the dashed green box in Panel (**f**). The middle image in Panel (**h**) refers to the region covered by the dashed yellow line in the image to its left. The inset contrast intensity profile (Panel **h**, right) refers to the three elements covered by this dashed yellow line.

same thickness of 180 μm were captured and are shown in Figs. 5b, 5c, respectively. The tetraSH-DVO2 lens was able to depict the shape of the steel mesh as well as the contour of the heat plate, whereas the PETA lens only captured a blurred shape of the steel mesh and failed to identify the heat plate. This comparison indicates that for LWIR imaging applications, the conventional resins used for 3D printing or UV curing are not suitable even with a thin thickness, but our multithiol-DVO resins offer clear advantages. Figure 5d shows a comparison of a tree branch captured under visible light (left) and LWIR light (right). Although the tree itself does not generate heat, the increased temperature was caused by the sun. This result demonstrated that optics fabricated using multithiol-DVO resin have the potential to be used for applications in more complex environments.

The self-fabricated mounting structure also enables easy tuning of the imaging distance (Fig. 5e), allowing for different field of view (FOV) to be achieved without requiring a change of lens. As demonstrated in Fig. 5f–h, the FOV gradually decreases as the imaging distance increases, resulting in an increase in imaging magnification and the ability to observe smaller objects with greater detail. In Fig. 5h, all the elements under group 3 on the USAF 1951 target can be seen under high magnification, indicating that the imaging resolution can reach 35 μm. This resolution is highly valuable for LWIR imaging applications, considering that the pixel size of the sensor is only $12 \times 12$ μm, and nearly reaches the limitation of the sensor.

## Monitoring reaction temperature change using a micro-reactor fabricated by IR transparent material

In addition to its use in the fabrication of IR optics, we explored the potential of our IR transparent thiol-ene resin for monitoring

temperature changes in micro-reactors through LWIR detection. The ability to monitor temperature changes during reactions in micro-reactors or microfluidic systems is critical for many applications[54]. Typically, conventional methods use thermocouples, thermistors, or other custom-bult sensors to measure the temperature at a specific location. These methods provides high-accuracy temperature measurements but have limitations measuring temperature change of a continous region. Robin G. Geitenbeek et al. reported using $NaYF_4$ nanoparticles for in situ temperature mapping in microfluidic systems through ratiometric bandshape luminescence thermometry[55]. The limitation for such strategy is that the nanoparticles need to be mixed into the solution. Herein, our IR transparent resin offers an alternative for monitoring temperature dynamics within reactors or channels over a broader scope, obviating the need for nanoparticles. The material's 3D printability further lends itself to customization of micro-reactors or microfluidic channels. To demonstrate this, we employed a stitch-free printing method to create a micro-reactor with three walls made of DUDMA and one wall made of tetraSH-DVO2, as shown in Fig. 6a. To demonstrate the temperature monitoring function, we positioned the tetraSH-DVO2 wall facing the LWIR camera and filled the micro-reactor with either 0.6 μL water or 0.6 μL 5 M HCl. We then added 0.6 μL of 5 M NaOH solution to the reactor, respectively. As shown in Fig. 6b, when there is only water in the reactor, the heat generated by the dilution of NaOH is not high enough to be detected. However, when 5 M NaOH was mixed with 5 M HCl, the acid-base neutralization released a larger amount of heat, which could be detected by our self-assembled LWIR camera (middle picture in Fig. 6c). When the 5 M NaOH was replaced with a NaOH particle, both the dissolving and neutralization generated much more energy in a shorter period, resulting in a brighter region in

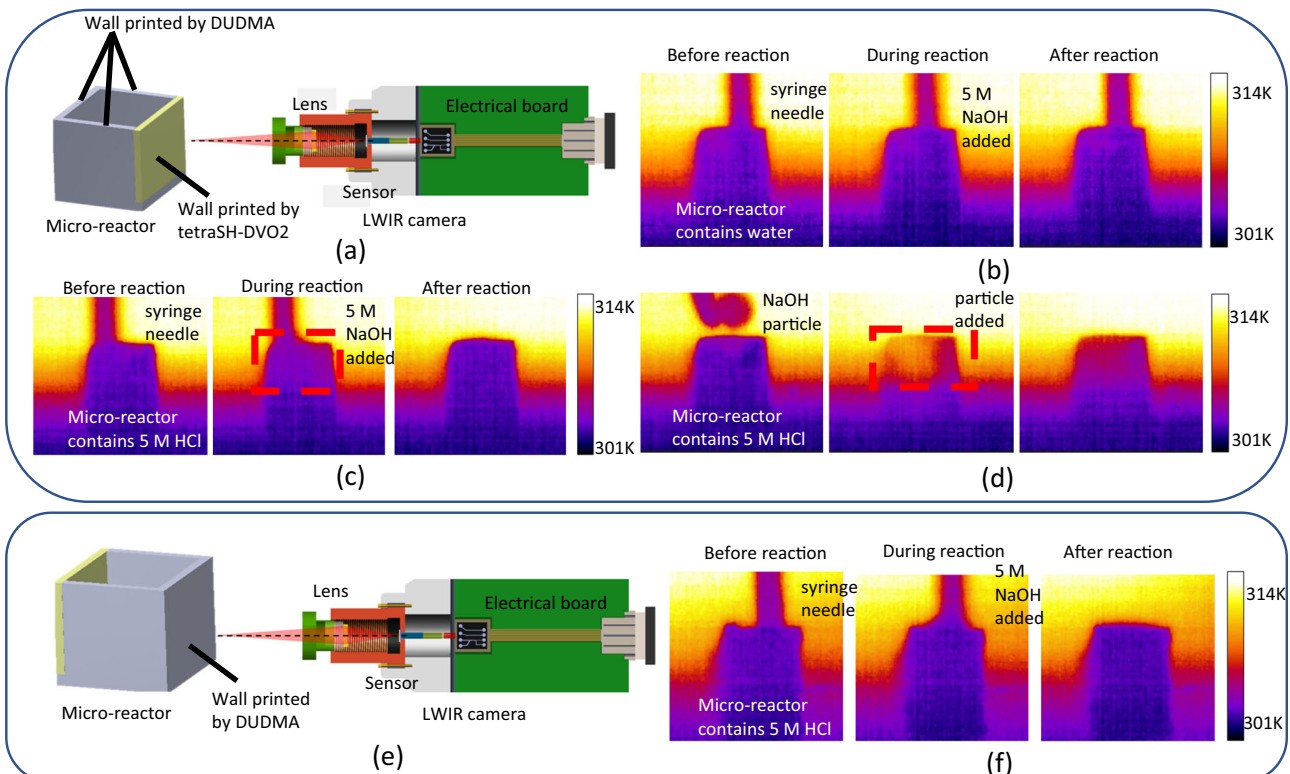

**Fig. 6 | Performance of 3D printed micro-reactor for temperature monitoring.** a Scheme of using an Long-wave infrared (LWIR) camera to monitor temperature change inside of a printed micro-reactor. The IR transparent wall faced the camera. The wall thickness is 100 μm. b The 5 M NaOH was mixed with DI water. The temperature change is not significant enough to be detected with the set up. c The 5 M NaOH was mixed with 5 M HCl. The reactor turned brighter in the LWIR range right

away after the NaOH was added. d A NaOH particle was added to the 5 M HCl. The reactor became much brighter in LWIR range. e The IR untransparent wall faced the camera. The wall thickness is 100 μm. f The temperature change inside the reactor cannot be detected when the diurethane dimethacrylate (DUDMA) wall faced the camera.

the LWIR image (Fig. 6d). Conversely, when the wall printed with DUDMA faced the LWIR camera (Fig. 6e), no significant changes were observed when NaOH was mixed with HCl (Fig. 6f), indicating that most of the IR signal was absorbed by the cured DUDMA wall (Supplementary Movie 1 is available for all examples shown here). Thus, we demonstrated that it is possible to monitor temperature changes through a thin layer of our multithiol-DVO material using an LWIR camera, providing greater flexibility in monitoring reactions in microreactors or microfluidic systems. It should be pointed out that the temperature monitoring sensitivity of these strategy may not be comparable to traditional thermalcouples and needs more exploration in future study.

## Discussion

In summary, we have successfully developed a UV-curable thiol-ene system with multithiols and divinyl oligomers only based on S, C, and H atoms with maximum transparency higher than 60% in the MWIR range and 20% in the LWIR range at the thickness of 500 μm. We have demonstrated that besides the functional groups, the hydrocarbon structure (chain length of $CH_2$ between S) also plays an important role affecting materials' LWIR transparency. The reported molecular design strategy ensures the resin shows impressive transparency in both the MWIR range and LWIR range compared to other commercial or reported UV-curable resin for photo-based additive manufacturing. Thermal and mechanical studies indicate that the monomer conversion and the mechanical properties of photo-cured resin can be improved by thermal post-curing process. We have demonstrated that optics with low surface roughness (<5 nm) and high surface accuracy (average error <1 μm) can be fabricated by both UV-assisted molding and photo-based 3D printing methods. High imaging resolution (<50 μm) was reached for both MWIR and LWIR imaging applications. IR transparent micro-reactor was also 3D printed as an example to show the potential of this material for temperature monitoring applications.

It is important to acknowledge that the current multi-thiol-DVO system still possesses limitations concerning sample thickness in comparison to certain S-rich polymers generated through inverse vulcanization. However, the 3D printability inherent in this system confers upon it a distinctive value in the efficient fabrication of thin IR optics as and when required.

Overall, our research presents a methodology for creating compact objects with infrared transparency, showcasing valuable implications for a wide array of industries and applications, including sensing, imaging, and chemical processing monitoring. The promising results yielded by our study suggest that our material holds the promise to bring about transformative effects within the realm of photo-based additive manufacturing and beyond, openning up possibilities for improving modern manufacturing techniques.

## Methods

### Materials

Epichlorohydrin, 2-mercaptoethanol, 5-Vinyl-2-norbornene, polybutadiene(90% 1,2-vinyl), 1,6-hexanedithiol, 1,8-octanedithiol, pentaerythritol tetraacrylate (PETA), diurethane dimethacrylate (DUDMA), pyrogallol, and 2,4-diethyl-9H-thioxanthen-9-one were purchased from Sigma-Aldrich. Bis(2-mercaptoethyl) sulfide and 1,10-Decanedithiol were purchased from TCI America. All the chemicals were used as received.

### Synthesis of 1-chloro-3-(hydroxyethylthio)-2-propanol (CHTEP)

The synthesis procedure was modified based on the procedure in previous literature[56]. Epichlorohydrin (10.00 g, 0.108 mol), borax (4.12 g, 0.011 mol), and DI water (50 mL) were mixed in a 250 mL round bottom flask. 2-mercaptoethanol (8.44 g, 0.108 mol) was then added to the flask dropwise with stirring at room temperature. The mixture became a homogenous solution gradually during the addition. This

solution was stirred at room temperature for 4 h. The solution was then extracted using chloroform (100 mL x 3). The combined organic layer was then washed with brine (50 mL) and dried using $MgSO_4$. The solvent was then removed using rotary evaporation to obtain a colorless and viscous oil (10.38 g, 56% yield) which was used in the next step without further purification. $^1$H NMR (500 MHz, $CDCl_3$) δ = 4.01-3.95 (m, 1H), 3.80 (t, 2H), 3.68-3.61 (m, 2H), 2.88-2.70 (m, 4H); $^{13}$C NMR (500 MHz, $CDCl_3$) δ = 70.63, 61.22, 47.91, 36.27, 35.92.

### Synthesis of tetraol

The synthesis procedure was modified based on the published literature[56]. In a 250 mL round bottom flask, ethanol (50 mL), NaOH (2.15 g, 0.054 mol), and bis(2-mercaptoethyl) sulfide (4.146 g, 0.0269 mol) were mixed and stirred for 10 min. After that, CHTEP (9.219 g, 0.0537 mol) was added dropwise. A white precipitate was formed during the addition. The suspension was stirred at room temperature overnight. After this period, 36% HCl (7.68 g) was added slowly to the suspension and all the precipitate was filtered off. The solvent in the obtained solution was then removed under vacuum to get a colorless and viscous oil (8.055 g, 70.8% yield) which was used in the next step without further purification. $^1$H NMR (500 MHz, MeOD-d6) δ = 3.88-3.81 (m, 2H), 3.69 (t, 4H), 2.84-2.64 (m, 20H); $^{13}$C NMR (500 MHz, MeOD-d6) δ = 70.84, 61.18, 37.58, 37.27, 34.75, 32.60, 31.77.

### Synthesis of tetrathiol (tetraSH)

The synthesis procedure follows the published literature[56]. Tetrol (8.055 g, 0.0191 mol), 36% HCl (17.9 g), and thiourea (8.1 g, 0.106 mol) were added to a 3-neck round bottom flask. The mixture was heated to 110 °C for 1 h. After this period, the solution was cooled to room temperature and the 50 wt% NaOH solution (17.88 g) was added under $N_2$ atmosphere. The mixture was then stirred at room temperature for 24 h. After that, the aqueous layer was extracted using toluene (100 mL x 4). The combined organic layer was then washed with 1 M HCl (100 mL), water (100 mL), brine (100 mL), and dried with $MgSO_4$. The toluene was then removed with reduced pressure to obtain a colorless oil (5.81 g, 62.4% yield). This tetrathiol was used without further purification. $^1$H NMR (500 MHz, $CDCl_3$) δ = 3.15-2.60 (m, 26H), 1.82-1.66 (m, 4H); 13 C NMR (101 MHz, $CDCl_3$) δ = 51.6, 48.9, 37.1, 36.1, 35.7, 28.8, 28.7, 28.2, 25.1, 24.9; LRMS (ESI): calculated $[M + Na]^+$: 508.87, found: 508.97.

### Synthesis of polythiol (polySH) using polybutadiene

To a 100 mL round bottom flask, polybutadiene (1,2 addition, 0.50 g, 0.0083 mol of vinyl group), 2,2′-thiodiethanethiol (1.27 g, 0.0083 mol, 1 equiv. to vinyl group), benzophenone (20 mg, 0.1 mmol) and 50 mL of THF were added. This solution was exposed to UV for 1.5 h with stirring at room temperature. After that, the solvent was removed using rotary evaporation and then a high vacuum with a cold trap filled with liquid $N_2$ to obtain a highly viscous colorless oil. $^1$H NMR and $^{13}$C NMR show that all the vinyl groups from polybutadiene were consumed. The synthesis procedure of tetraSH and polySH was shown in Fig. 7.

### Synthesis of divinyl oligomers (DVO)

To a 25 mL round bottom flask, 5-vinyl-2-norbornene (2.00 g, 0.017 mol,) was added. This flask was then purged with $N_2$ for 5 min. Then, Bis(2-mercaptoethyl) sulfide (0.008 mol) was added dropwise under $N_2$. The solution was allowed to be stirred at room temperature for 24 hours. After that period, the vinyl concentration of the final liquid product was determined using NMR with MEHQ as a calibrator. The diene oligomers prepared using other dithiols (1,6-hexanedithiol, 1,8-octanedithiol, and 1,10-decanedithiol) were prepared using a similar method.

### Titration of thiol concentration in tetraSH and polySH

The titration was done following previous literature[57]. In general, in a 250 mL Erlenmeyer flask, around 0.15 g polythiol and 15 mL pyridine

**Fig. 7 | Steps of synthesizing multithiols. a** Synthesis of 1-chloro-3-(hydroxyethylthio)-2-propanol (CHTEP). **b** Synthesis of tetraol. **c** Synthesis of tetrathiol. **d** Synthesis of polythiol.

were mixed together to form a solution. 5 mL silver nitrate solution (0.4 M) was then added to this solution to form a yellow suspension. This suspension was allowed to sit for 5 min. After that period, 100 mL $H_2O$ and 1 drop of phenolphthalein (1% solution) were added to the suspension. This solution was then titrated using 0.1 M NaOH to light pink as the endpoint. Each batch of polythiol was titrated individually to get the accurate thiol concentration.

### Preparation UV-curable thiol-ene resin using tetraSH (or polySH) and DVO

In general, the UV curable thiol-ene resin were prepared by mixing diene oligomer and tetraSH (or polySH) together based on the vinyl concentration of diene oligomers and the thiol concentration of tetraSH (or polySH). TetraSH was firstly mixed with 2,4-diethyl-9H-thioxanthen-9-one (initiator, 2 wt% calculated based on the total mass of TetraSH and DVO) and pyrogallol (inhibitor, 0.1 wt% calculated based on the total mass of TetraSH and DVO). DVO was then added to the vial. This mixture was stirred until a homogenous resin was formed. The resin can be used for either UV curing or two-photon printing directly.

### Prepare UV-curable acrylate and methacrylate resin for comparison

To prepare the comparison resin containing acrylate and methacrylate groups, pentaerythritol tetraacrylate or diurethane dimethacrylate were mixed with initiator (2 wt% to resin), respectively. The resin can be cured under UV irradiation.

### Computation of IR spectra of model compounds

The calculation of IR spectra of model compounds was conducted using Gaussian 16. For each molecule, the energy was firstly optimized using B3LYP/6-31*. Then, the frequency calculation was done with the energy-optimized configuration using the B3LYP/6-31* set. The predicted IR spectra of the gas phase can be obtained from the frequency calculation output using GaussSum with a scaling factor of 0.98.

### Two-photon printing of multithiol-DVO resin

The printing system contains a 780 nm femtosecond fiber laser with 150 fs pulse, 77 MHz, and a maximum power of 130 mW. The full-width half maximum (FHWM) of the beam was 5 mm, 83% filling the objective (NA = 0.6). The component was printed with a 1.17 nJ pulse energy and 16900 μm/s on the Sodium-chloride substrate. After printing, the uncured resin was washed using THF for 15 mins and post-cured under UV exposure for ten mins.

### Post-curing of UV-cured thiol-ene resin with heat

When thermal post-curing of the UV-cured sample is needed, the UV-cured samples were placed in a sealed oven. Nitrogen was purged for 15 min before increasing the temperature. After that

period, the samples were heated to 170 °C and kept for 12 h in $N_2$ atmosphere, and cooled to room temperature before they were removed from the oven.

### Fabrication of IR transparent objects by molding and 3D printing

To demonstrate the imaging capabilities of the developed IR transparent resin, we utilized both molding and two-photon direct ink writing methods to fabricate optics for testing imaging performance in the MWIR and LWIR ranges. The refractive index (RI) of both tetraSH-DVO2 and polySH-DVO2 are measured before the lens design. The tetraSH-DVO2 shows a slightly higher RI (~1.55 from MWIR to LWIR) compared to polySH-DVO2 (~1.54 from MWIR to LWIR) (Supplementary Fig. 16). This is mainly because the S ratio in tetraSH (59 wt%) is higher than it in polySH (46 wt%). The molding process (Supplementary Fig. 17a) involved curing the resin directly on a NaCl substrate using a glass mold (~180 μm Sagitta length). The high manufacturing efficiency of this process was evident in the quick curing of the millimeter-sized lens (Supplementary Fig. 17b) within a minute. After UV curing, the molded lens was thermally cured together with the mold and the NaCl substrate at 170 °C for 12 h. The molded lens exhibited a smooth surface with a surface roughness of only 4.9 nm (Supplementary Fig. 18a). For the two-photon printing process, a 3 × 3 lens array (Supplementary Fig. 17e) with lenses' diameter of 450 μm and a thickness of 25 μm was directly printed onto a NaCl substrate equipped with the pre-printed polymer mask, showcasing the printing quality that reached a surface roughness as low as 4.3 nm (Supplementary Fig. 18b). The lens array was not thermally cured before imaging test. Supplementary Fig. 17d shows the setup utilized for printing process. Supplementary Figs. 17c, f shows the cross-section shape of the lens in Supplementary Fig. 16b and one singlet lens in the lens array in Supplementary Fig. 17e, respectively. The results indicate that the designed shape can be well-maintained by either the molding method or the 3D printing method. It should be noted that we found out the printed lens array has some swell behavior during the washing process using THF, which could be due to the relatively low monomer conversion after 3D printing compared to commercial resin. This could potentially be solved by using a more efficient initiator or increasing the laser power during printing. Another problem that has been noticed is that although both the tetraSH-DVO2 and polySH-DVO2 resins are 3D printable, the polySH-DVO2 resin has even slower curing efficiency and lower printing quality, making it difficult to precisely control the surface shape of printed optics. This result agrees with the results of the thiol conversion that polySH-DVO2 reached around 50% thiol conversion during the first 10 min's UV exposure Therefore, only the tetraSH-DVO2 resin was used for 3D printing optics.

## Fabrication of micro-reactor with stitch-free strategy

To fabricate the micro-reactor with 3 walls made of DUDMA and 1 wall made of tetraSH-DVO2, we first used DUDMA with an initiator to print 3 walls and the bottom structure by two-photon polymerization. The wall thickness was set as 100 µm. After washing off the uncured DUDMA, one drop of tetraSH-DVO2 resin was dropped to surround the printed structure. The focal point of the laser was carefully located in the desired position. The IR transparent wall was then printed with a thickness of 100 µm. The printed micro-reactor has a width and length of 1.1 mm and a height of 900 µm (Supplementary Fig. 19). It can be observed that the printing resolution of tetraSH-DVO is lower than the printing resolution of DUDMA.

## Fabrication of structure with micro-channels using tetraSH-DVO2

To explore the printing capability for the fabrication of thin channel structures using tetraSH-DVO2 resin, we designed and printed structures that contained micro-channels. It was found out that the smallest diameter of the micro-channels printed using tetraSH-DVO2 is around 10 µm (Supplementary Fig. 20). Smaller channel diameter may cause a merge phenomenon which indicates that the printing resolution of current tetraSH-DVO resin is limited to several micrometers. However, this resolution is enough for most MWIR and LWIR applications since they all utilize IR with wavelengths higher than 3 µm.

## Characterization

IR spectra were obtained with a Thermo Scientific Nicolet iS50R. The transmission spectra of different samples were measured with the transmission mode. And the conversions during material curing were measured with the ATR mode. Nuclear magnetic resonance (NMR) was obtained using a Bruker DRX 500 MHz. Scanning Electron Microscope (SEM) images were taken using FEI Inspect Scanning Electron Microscope. The surface profile was measured by Zygo Newview 8300 white light interference microscope. The thermogravimetric analysis (TGA) was carried out with TA TGA 5500 (10 °C/min, 35 °C to 800 °C) and the differential scanning calorimetry (DSC) was carried using TA DSC 2500 (20 °C/min, −60 °C to 200 °C).

The authors affirm that human research participants [or their parents/guardians] provided informed consent for publication of the images in Fig. 4.

## Prepare the thin plate for FTIR transmission test

For UV curable resin, on a KBr sample card (Real Crystal®), the prepared UV curable resin was dropped to full fill the space above the KBr crystal. To minimize the potential deformation caused by shrinkage during the curing, a glass slide coated with PDMS was covered above the thiol-ene resin. The resin was then cured using OmniCure S2000 (~3 W/cm² source power, 10 cm distance between resin and light source, 10 min exposure time) to obtain a thin window for FTIR transmission test. The thickness of each window is between 500 to 550 µm.

For LDPE and HDPE, the polymer pallets were placed on a PDMS coated glass slide. A heating plate was placed below the glass slide to melt the PE polymer. A KBr sample card was then covered and pressed above the PE polymer to form a LDPE (or HDPE) window with a thickness between 500 and 550 µm.

## Measure conversion of thiol and alkene during UV curing

To measure the thiol (alkene) conversion versus UV exposure time, a series of thin samples with ~1 mm thick were cured under UV with the same curing conditions (OmniCure S2000, ~3 W/cm2 source power, 10 cm distance between resin and light source) except for curing time (from 20 s to 20 min for difference samples). The cured samples were measured by FTIR-ATR to obtain the spectra. To calculate and compare the conversion, the obtained spectra were normalized based on the alkane peaks (~2900 cm⁻¹). The conversion of thiols and alkene was then calculated based on the integration change of the peak near 2500 cm⁻¹ and the peak between 3000 and 3100 cm⁻¹. It is noteworthy that it is difficult to guarantee everything in each sample is exactly the same. Therefore, the conversion in the stable region (samples cured from 1 min to 10 min) fluctuated. But the stable trend can still be observed indicating the conversion is stable within this time interval. For a long time post UV curing, the samples cured for 20 min by Omnicure were transferred to a Formlabs curing machine and cured for 12 h at room temperature. After that, the conversion was calculated based on the results from FTIR-ATR.

## Prepare the mechanical testing coupons using thiol-ene resin for tensile test

ASTM D638 type IV coupons were firstly 3D printed. Then, the printed coupons were used as a positive mold to fabricate a negative mold using PDMS.

With the PDMS negative mold, thiol-ene resin were poured into the mold and exposed under UV to finally obtain the ASTM D638 type IV coupons for tensile test

## Tensile testing

The tensile testing was carried out using Instron tensile machine with tensile mode (1 mm/min to break). When the cyclic tensile testing was conducted, the samples were stretched to 30% strain compared to original length and then gradually pushed back to original length with the rate of 1 mm/min. For both tetraSH-DVO2 and polySH-DVO2 with and without thermal post curing, 6 samples were tested in tensile testing.

## Prepare samples using thiol-ene resin for DMA test

To prepare the samples for DMA test with compression mode, the thiol-ene resin was filled in a hollow cylinder with 3 mm inner diameter and 1 mm thickness between two glass slides coated with PDMS. The resin was then cured using OmniCure S2000 (~3 W/cm² source power, 10 cm distance between resin and light source, 10 min exposure time).

## Dynamic mechanical analysis

The DMA test was carried out using DMA 242 E under compression model. The temperature was set from −50 °C to 200 °C with an increasing rate of 3 °C/min.

## Measurement of refractive index

A TFProbe InfraRed Spectroscopic Ellipsometry (IRSE) tool (made by Angstrom Sun Technologies Inc.) is used to characterize the refractive index of materials in IR range. To characterize the material in the infrared range (350–7400 cm^-1), an infrared light source input is used inside a Michelson interferometer setup. A liquid nitrogen cooled MCT detector is used to receive the IR signal after it passes through the ellipsometer's polarizing optics and reflected off the test sample. The IRSE measurement was taken at an angle of incidence (AOI) of 70 degrees.

The measurement was taken and analyzed using the TFProbe 3.3 software. A bare substrate and the sample of interest were both measured, and the analysis was done by creating a model to fit onto the measured data. A dispersion type fitting algorithm was used to characterize and calculate both the thickness and optical constants (NK) of the samples. For improved measurement accuracy, a bare substrate of the sample was first measured and modeled to obtain the NK constants and subsequently used within the model for the sample of interest.

# Data availability

The authors declare that the data supporting the findings of this study are available within the paper and its Supplementary Information files. Should any raw data files be needed in another format they are

available from the corresponding author upon request. Source data are provided with this paper.

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

## Acknowledgements

This work was supported by National Cancer Institute R21CA268190. The authors wish to express their gratitude to Dr. Ronald G. Driggers, Patrick Leslie, and Lindsey Wiley for their assistance with MWIR imaging experiments.

## Author contributions

P.Y. and Z.H. contributed equally to this work. P.Y., Z.H., D.A.L. and R.L. conceived the idea and designed the study. P.Y. designed, prepared, and characterized the materials. Z.H. designed the optics, performed printing experiments, and designed the imaging experiments. P.Y., Z.H., D.A.L. and R.L. analyzed and interpreted the result, and wrote the manuscript.

## Competing interests

The authors declare no competing interests.
