## [Peer Review File · Nature Communications]

UV-Curable Thiol-ene System for Broadband Infrared Transparent ObjectsREVIEWER COMMENTS

Reviewer #1 (Remarks to the Author):

The author has successfully developed a novel UV photocurable material system exhibiting remarkable transparency in the MWIR and LWIR spectra. The samples were subjected to rigorous testing to evaluate their refractive index, thermal properties, and mechanical characteristics. The study highlights several promising applications, such as the production of high-resolution IR optics lenses for compact IR imaging systems, as well as the fabrication of micro-reactors for temperature monitoring purposes. Overall, the paper is well-written, and I highly recommend its acceptance after revision. I would like to provide the following comments for consideration:

1. Further literature review is necessary to explore the thiol-ene system in greater depth, focusing on its advantages, benefits, and wide-ranging applications.
2. It would be valuable to compare the proposed material system's LWIR/MWIR transmission capabilities with those of traditional materials to assess its performance and potential advantages.
3. Could you provide information on any post-treatment methods employed for the 3D printed microlens? Understanding the post-treatment process will aid in comprehending the overall fabrication and performance of the microlens.
4. It is recommended to adjust the arrangement of sub-images in Figure 4, 5, and 6 as the current sub-image sizes, particularly in Figure 5 (e-h), are too small to adequately convey the details. Enhancing the readability and visibility of the sub-images will significantly improve the presentation quality.
5. Please clarify the benefit of 3D printed micro-reactor on the monitor temperature change compared with traditional technologies.
6. The "Results and Discussion" section lacks a comprehensive description of the microchannels and their intended application. It is advisable to provide additional details regarding the microchannels, their purpose, and their potential applications to provide a more comprehensive understanding of their significance in the research findings.

Reviewer #2 (Remarks to the Author):

This manuscript reported a thiol-ene photo-curable liquid material for the fabrication of IR transparent objects, where the multithiols and divinyl oligomers (DVO) were designed to contain only C, H, and S atoms. The resulting curing materials is transparent in the range of mid-wave IR (MWIR), and long-wave IR (LWIR). The main points in this work is the new synthesized tetraSH and polySH. In fact, the similar strategy of thiol-ene photo-curable materials for IR lens containing the same atom C, H, and S was reported in other work by using the commercial triSH. Therefore, the novelty of this work, in my opinions, can not reach the publishment in Nat Commun. One more comment: It is better to provide the refractive index of the cured materials in the text.

Reviewer #3 (Remarks to the Author):

Manuscript ID: NCOMMS-23-23056

Title: " UV-Curable Thiol-ene System for Broadband Infrared Transparent Objects"

Author(s): Piaoran Ye, Zhihan Hong, Douglas A. Loy, Rongguang Liang

Content: The authors present the synthesis of a long-wavelength infrared (LWIR) transmitting polymer consisting of sulfur and several newly synthesized compounds. The synthesis method utilized UV-initiated thiol-ene coupling chemistry. The properties of the material were analyzed using numerous optical, thermal and mechanical techniques. Computational analyses were also provided. The work has potential in optical sensing applications.

Impact/Novelty: From this reviewer's perspective, the impact and novelty of this effort are both moderate. The impact of this work is that it expands the growing field of sulfur-based polymers that have optical transparency from the visible into the LWIR. The novelty of this work is the development of a UV curable system in which the product is a material that transmits in the LWIR. The use of thiol-ene chemistry to accomplish this (rather than the seemingly ubiquitous inverse vulcanization method) is also unique. That said, the particulars regarding the optical properties of the polymer are different than other reports, but not particularly superior. For example, Figure 1 shows a transmission plot of the compounds synthesized for this study. Although there is usable transmission (up to ~20%) in the LWIR of the plots using the synthesized compounds, the values are not a major improvement over those that have been published by Kleine et al. (doi.org/10.1002/anie.201910856), or Boyd et al. (doi.org/10.1021/acsmacrolett.8b00923). This is particularly true when considering that the work of both Kleine and Boyd used samples greater than 1.5 mm in thickness, while the present submission uses samples that are 0.5 mm in thickness. The authors even point out in the text describing Figure 2 that the LWIR transparency was minor for samples that were 0.9 mm thick. Therefore, the claim of having significantly improved LWIR transparency is overstated, and renders these results different, but not remarkable.

Grammar/Language: The grammar and language are appropriate.

Additional Comments: The authors made multiple comments pointing out that the use of aromatic reagents is a hindrance to LWIR transparency. A recent preprint on the topic of inverse vulcanization demonstrates that this is not the case (see: [10.26434/chemrxiv-2023-qjkn](https://doi.org/10.26434/chemrxiv-2023-qjkn)). In the preprint, isomers of divinylbenzene are all shown to transmit (~10%) in the LWIR. Whether or not the preprint details are an exception, as opposed to a norm, is not yet clear. Nevertheless, the authors of this submission should update the language in the manuscript with respect to the preprint mentioned.

Thiol-ene and Thiol-yne reactions are known to proceed via thermal initiation (as well as other methods such as UV initiation). For references, see doi.org/10.1002/macp.200900442 and doi.org/10.1021/acsomega.8b00319. Therefore, the fact that the authors see an increase in the storage modulus as the temperature increases suggests that the polymer is not fully cured under UV light (as the authors point out). Did the authors consider doing both a UV post-cure of the polymers (ideally under

inert atmosphere) as well as a comparative thermal post-cure? This question arises because one route might lead to more curing than the other, thus leading to a more stable product.

For the data in Figure 3, including a table with the specific values (e.g. T_g, stress/strain) would be very helpful.

Recommendation: The work detailed in this effort is well done, and there is definitely value in the work presented. Specifically, the use of thiol-ene chemistry to develop LWIR-transmissive materials is noteworthy. Also, expanding the field of materials (and methods to develop materials) that possess LWIR transmission capabilities is valuable. However, there are a number of published papers in which similar results have been obtained (though not using thiol-ene chemistry). That said, this reviewer would ask the authors to consider adjusting the language throughout the manuscript to better reflect this effort's place within the broader context of recent work detailing the synthesis of LWIR-transmissive polymers.

Responses to Reviewers' Comments

Reviewer 1:

We sincerely appreciate the comments that help improve our manuscript. Below are our responses to each comment.

1. Further literature review is necessary to explore the thiol-ene system in greater depth, focusing on its advantages, benefits, and wide-ranging applications.

Response: Thanks for giving comments to help enrich the instruction and background of our manuscript. We did more literature search and added the content to introduce more about thiol-ene system in the introduction section:

“Thiol-ene click reaction is famous for its high efficiency.²⁴ This reaction mechanism, operating through a free radical process, offers versatility by being triggered through either thermal or photo initiation, both with or without initiators²⁵⁻²⁸ Such feature not only makes the thiol-ene system as a powerful tool in polymer synthesis and bio-synthesis,³⁰ but also makes it widely used in the material science including coating,³¹ molding,³² and additive manufacturing.³³⁻³⁵ Notably, the majority of thiol-ene materials exhibit favorable transparency within the visible light range, rendering them a prime choice for fabricating optics characterized by diverse mechanical and optical properties.³⁶⁻³⁹

However, most thiol-ene systems are unsuitable for infrared imaging applications, especially within the LWIR region. This is primarily due to the presence of molecular structures, such as ether and carbonyl groups, which lead to IR absorption across both MWIR and LWIR spectra, resulting in diminished transparency.¹⁹ A noteworthy endeavor by Yang Qiu et al. involves imprinting a high refractive index Fresnel Lens structure onto silicon or quartz surfaces using a trithiol monomer.⁴⁰ This system's absence of oxygen allows it to exhibit transparency in both the MWIR (200 μm thickness, 62% transparency) and LWIR (200 μm thickness, 24% transparency) regions. This work utilized a low molecular weight trithiol with allyl, styryl and propargyl modified thiols. Additionally, the utilization of photo-based additive manufacturing to fabricate optics transparent in both MWIR and LWIR regions has yet to be explored.”

2. It would be valuable to compare the proposed material system's LWIR/MWIR transmission capabilities with those of traditional materials to assess its performance and potential advantages.

Response: Thank you for your comment. We have attached a table (Table S1) in the supporting document to compare our materials with other reported systems. In this table, polyethylene and Ge are also listed as references for IR transparency comparison. We didn't include chalcogenide glasses in this table since the formula of such glasses can be very different (typically, they have transparency in both LWIR and MWIR higher than 40% with a thickness of 1mm or higher). And this manuscript focuses more on the organic system. As for commercial polymers for IR applications, polyethylene, or similar polyolefin polymers, might be the most affordable one that has

been widely used. There are some other options such as Poly-View™ UL 746. Unfortunately, we haven't found public transparency data on this material.

3. Could you provide information on any post-treatment methods employed for the 3D printed microlens? Understanding the post-treatment process will aid in comprehending the overall fabrication and performance of the microlens.

Response: We apologize for this. For the molded lens, we do the thermal post-curing in nitrogen with the mold together. After that, we do the demold process to remove the mold. For tetraSH, we didn't observe an obvious shrinkage after this process. But if the thermal treatment is made to a free-stand sample without attached mold and substrate, we can observe a shrinkage (~5%). The shrinkage for polySH is a little bit larger, reaching ~12%.

For the two-photon printed lens array, we didn't do a thermal post treatment. When doing the printing, we attach the mask containing 9 holes directly on the NaCl substrate, and print each singlet micro-lens inside of each hole. To make sure that the mask can block unwanted light, we didn't do the thermal treatment to avoid possible shrinkage of the printed IR lens. Based on the IR transmission test, the thermal post curing slightly increased the transmission in the LWIR region (see the revised Fig. S14c and d). In our imaging experiments, this difference didn't bring significant improvement to imaging quality. However, if necessary, the 3D printed lens can be thermally treated for improved mechanical and thermal properties as described in the manuscript.

We have added relevant content in the Method section to make it clear.

“After UV curing, the molded lens was thermally cured together with the mold and the NaCl substrate at 170 °C for 12h.”

“The lens array was not thermally cured before the imaging test.”

Also the content in the main text:

“We observed that the thermal post-curing slightly increased the transmission in the LWIR region (Fig. S14c and d), but this difference is not significant enough to improve the imaging quality in the later imaging test.”

4. It is recommended to adjust the arrangement of sub-images in Figure 4, 5, and 6 as the current sub-image sizes, particularly in Figure 5 (e-h), are too small to adequately convey the details. Enhancing the readability and visibility of the sub-images will significantly improve the presentation quality.

Response: Thanks for the comments. We have enlarged the sub-images as well as the text in Figure 4, 5, and 6.

5. Please clarify the benefit of 3D printed micro-reactor on the monitor temperature change compared with traditional technologies.

Response: Thanks for the comment. We added a discussion to show why we would like to explore the possibility of using our material to fabricate micro-reactor to monitor the temperature change.

“These methods provide high-accuracy temperature measurements but have limitations measuring temperature change of a continuous region. Robin G. Geitenbeek et al. reported using NaYF₄ nanoparticles for in situ temperature mapping in microfluidic systems through ratiometric bandshape luminescence thermometry.⁵⁵ The limitation of such a strategy is that the nanoparticles need to be mixed into the solution. Herein, our IR transparent resin offers an alternative for monitoring temperature dynamics within reactors or channels over a broader scope, obviating the need for nanoparticles. The material's 3D printability further lends itself to customization of micro-reactors or microfluidic channels.”

Besides, we would also like to point out that the temperature monitoring experiments in this manuscript are an early stage principle proof. Therefore, we also added a sentence to clarify it.

“It should be pointed out that the temperature monitoring sensitivity of these strategy may not be comparable to traditional thermocouples and needs more exploration in future study.”

6. The "Results and Discussion" section lacks a comprehensive description of the microchannels and their intended application. It is advisable to provide additional details regarding the microchannels, their purpose, and their potential applications to provide a more comprehensive understanding of their significance in the research findings.

Response: We are grateful for the opportunity to enhance the clarity and depth of our "Results and Discussion" section. Originally, our objective in fabricating this micro-channel structure was to explore the spatial printing resolution of our multi-thiol-DVO system. Given that we briefly touched upon micro-reactors/micro-fluidic systems in the preceding paragraph of the Results section, we opted to print this micro-channel structure inspired by 3D micro integrated fluid system. However, due to the lack of convenient resources for building a comprehensive micro-fluidic system, we were unable to design and test a functional micro-channel for use in a micro-fluidic/reaction setup. As a result, we intentionally minimized our discussion on this topic within the main text, aiming to avoid undue emphasis on this aspect of our work.

Having taken your feedback into account, we acknowledge the significance of your insights in enhancing the depth of our manuscript. Consequently, we have included a discussion on the micro-channels in the supplementary material. Please see “Discussion on 3D printing of micro-channels” in the supplementary material:

“To prove the 3D printability as well as to test the printing resolution, we used tetraSH-DVO2 to print a multi-channel structure, which was inspired by 3D micro integrated fluid system,¹ as shown in Fig. S20.

Nowadays, most microfluidic devices are limited to two-dimensional microchannels because they were manufactured by photolithographic patterning on glass or silicon

substrates and the subsequent bonding process.² On the other side, 3D micro integrated fluid system is a concept that has been developed since 1990s, which contains 3D-structured microchannels and was originally developed for bio-synthetic and bio-sensing applications.¹ Such a strategy allows a more compact design and can integrate many functions into one microfluidic system. The similar concept has been applied to fabricate integrated microfluidic or microchannel systems that has other functions including micro-cooling system,^{3, 4} microsphere generation,⁵ fluidic coupler (mixer),⁶ etc. Regarding the fabrication method, 3D printing has its unique advantage in fabricating complex 3D structures and has been reported to fabricate 3D integrated microfluidics or microchannels.⁷⁻⁹

In this study, we present the fabrication of a micro-module incorporating microchannels using our tetraSH-DVO2 material. The resulting cubic structure contains 12 x 14 microchannels, each with an approximate diameter of 10 μm (Figure S20). It's worth noting that a minor flaw in printing quality is observed in the upper right corner of the structure. This phenomenon can be attributed to its proximity to the boundary of the objective's field of view during the printing process. Our investigations reveal that this diameter represents the lower threshold achievable for micro-channels when utilizing the tetraSH-DVO2 resin. While this resolution falls below that of most TPP 3D printing methods, and there's a possibility of merging in channels with smaller diameters, this resolution limitation of the tetraSH-DVO2 resin is still suitable for numerous MWIR and LWIR applications. These applications predominantly involve IR with wavelengths surpassing 3 μm . In future work, we intend to integrate the 3D structured microchannels with IR signal monitoring. This undertaking will necessitate proper design of the structure and a comprehensive optimization of both the resin composition and printing parameters.”

Reviewer 2:

This manuscript reported a thiol-ene photo-curable liquid material for the fabrication of IR transparent objects, where the multithiols and divinyl oligomers (DVO) were designed to contain only C, H, and S atoms. The resulting curing materials is transparent in the range of mid-wave IR (MWIR), and long-wave IR (LWIR). The main points in this work is the new synthesized tetraSH and polySH. In fact, the similar strategy of thiol-ene photo-curable materials for IR lens containing the same atom C, H, and S was reported in other work by using the commercial triSH. Therefore, the novelty of this work, in my opinions, can not reach the publishment in Nat Commun. One more comment: It is better to provide the refractive index of the cured materials in the text.

Response: We are grateful for the constructive feedback that has guided the enhancement of our manuscript. We identified a paper (<https://doi.org/10.1002/macp.202100311>) that aligns with the reviewer's remarks. We sincerely regret the oversight of this paper and have now incorporated it as a reference in the Introduction section, accompanied by a discussion: "A noteworthy endeavor by Yang Qiu et al. involves imprinting a high refractive index Fresnel Lens structure onto silicon or quartz surfaces using a trithiol monomer.⁴⁰ This system's absence of oxygen allows it to exhibit transparency in both the MWIR (200 μm thickness, 62% transparency) and LWIR (200 μm thickness, 24% transparency) regions. This work utilized a low molecular weight trithiol with allyl, styryl and propargyl modified thiols. Additionally, the utilization of photo-based additive manufacturing to fabricate optics transparent in both MWIR and LWIR regions has yet to be explored."

As highlighted by the Reviewer, we acknowledge the superficial similarity between the structures of one of our multi-thiol systems and the reported triSH system. However, we firmly believe that our research offers distinctive contributions and value that extend the advancement of transparent optics within the broadband IR range. Here are key points of differentiation that set our manuscript apart:

1. **Molecule Choice and Design:** Our Results section not only proposes the enhancement of IR transparency by employing C, H, and S-only molecules but also delves into the profound impact of the varying CH₂ bridge lengths between S atoms within the initial material. This aspect significantly influences LWIR transparency, a subject explored through empirical experimentation and computational insights. We think it is important since our work focused more on the improvement of the IR transparency rather than the improvement of refractive index. It is a more detailed study about how the molecules' structure affects LWIR transparency.
2. **Expanded Scope of Systems:** While the aforementioned paper employs triSH in a small molecule context, our work not only investigates tetraSH as a small molecule (or oligomer) system but also examines polySH as a polymer system. This widened scope introduces heightened flexibility for fabrication since these systems encompass viscosities ranging from low to high. Furthermore, polySH allows macromolecular structures to be incorporated into the thermoset network, unlike the small molecule precursors in the Qiu's paper, that should provide

enhanced mechanical properties. Another significant difference was our use of the vinylnorbornene to synthesize dialkenyl comonomers as opposed to Qiu's allyl, styryl and propargyl modified thiols.

3. **Exploration of TPP for IR Optics:** Our manuscript pioneers the application of TPP 3D printing for crafting thin IR optics using the multi-thiol-DVO system. In contrast, the aforementioned manuscript uses photocuring through a mold. Notably, TPP as a method for fabricating organic objects transparent to both MWIR and LWIR has yet to be reported.
4. **Comprehensive Thickness Study:** Our research conducts a more in-depth, meticulous analysis of the influence of sample thickness on IR transparency within both MWIR and LWIR ranges. By varying thickness and introducing C=O and C-O bonds, we provide invaluable insights into our multi-thiol system's performance under diverse conditions. This depth of understanding emphasizes our dedication to comprehending the intricate relationship between material composition, thickness, and transparency.
5. **Thermal and Mechanical Property Insights:** Another significant difference is that our investigation uncovered the necessity of post-thermal treatment to stabilize the multi-thiol system's thermal and mechanical properties. This aspect holds great significance, particularly for optical applications, where stability is imperative for optimal performance. In addition, we demonstrated that the polySH-DVO2 system shows a much higher stress after post-curing compared the tetraSH, showing an advantage of polymeric system compared to small molecule system.
6. **Detailed Performance Evaluation:** In our manuscript, we extensively evaluate the efficacy of IR optics produced using the multi-thiol system via both molding and TPP 3D printing. We showcase the precision and low surface roughness achieved through both techniques. Furthermore, our study undertakes a rigorous comparison of our manufactured lenses to commercial counterparts, while also investigating the inherent imaging resolution limits. These comprehensive analyses are pivotal in the context of practical IR imaging applications.

By highlighting these unique characteristics, we underscore the innovative contributions of our study, emphasizing its importance in the field of transparent optics for broadband IR applications.

Regarding the last comment about refractive index: the RI of both cured materials for the IR region has already been addressed in the Materials section of the manuscript (please see page 24, **Fabricate IR transparent objects by molding and 3D printing**).

Reviewer 3:

We acknowledge their insightful comments that help improve this manuscript. Please see our response point by point showing below.

1. Therefore, the claim of having significantly improved LWIR transparency is overstated, and renders these results different, but not remarkable.

Response: Thanks for pointing this out. We carefully checked the manuscript and revised the statement that was in the last paragraph. We also added a paragraph in the last section to clarify the limitation of current work.

“It is important to acknowledge that the current multi-thiol-DVO system still possesses limitations concerning sample thickness in comparison to certain S-rich polymers generated through inverse vulcanization. However, the 3D printability inherent in this system confers upon it a distinctive value in the efficient fabrication of thin IR optics as and when required.

Overall, our research presents a methodology for creating compact objects with infrared transparency, showcasing valuable implications for a wide array of industries and applications, including sensing, imaging, and chemical processing monitoring. The promising results yielded by our study suggest that our material holds the promise to bring about transformative effects within the realm of photo-based additive manufacturing and beyond, opening up possibilities for improving modern manufacturing techniques.”

2. The authors made multiple comments pointing out that the use of aromatic reagents is a hindrance to LWIR transparency. A recent preprint on the topic of inverse vulcanization demonstrates that this is not the case (see: [10.26434/chemrxiv-2023-qqjkn](https://doi.org/10.26434/chemrxiv-2023-qqjkn)). In the preprint, isomers of divinylbenzene are all shown to transmit (~10%) in the LWIR. Whether or not the preprint details are an exception, as opposed to a norm, is not yet clear. Nevertheless, the authors of this submission should update the language in the manuscript with respect to the preprint mentioned.

Response: We greatly value the insightful discussion regarding the impact of aromatic rings on LWIR transparency. Through further exploration of relevant literature, including the preprint provided by the Reviewer, we've discerned additional factors beyond the presence of the aromatic ring that plays a substantial role. Notably, the symmetry of the ring structure emerges as a pivotal consideration in determining LWIR transparency. It has come to our attention that certain materials, even when incorporating aromatic rings, possess the potential to exhibit high transparency within the LWIR region. This observation is exemplified by the use of DVB as opposed to DIB, wherein the material's LWIR transparency is heightened due to the influence of the methyl group.

Furthermore, in our quest for deeper insights, we encountered a recent paper (<https://doi.org/10.1038/s41467-023-38398-5>) that sheds light on an intriguing phenomenon. This study demonstrates that despite the presence of aromatic rings within the structure, the material's tri-functionalized, highly symmetric configuration yields impressive LWIR transparency results. Specifically, the material achieves over 70% transparency in the LWIR range with 80wt% S content, and over 40%

transparency with 50wt% S content. This finding accentuates the profound interplay between structural symmetry, composition, and transparency in the LWIR realm. Therefore, we have added content to instruction section:

“Efforts to enhance LWIR transparency, even in the presence of aromatic rings, have led to significant strides in recent research. Miyeon Lee et al. demonstrated the remarkable potential of using symmetric benzenetrithiol in conjunction with elemental sulfur, achieving peak transparency exceeding 70% in 1mm thick samples.¹⁶ Darryl Boyd et al. explored the influence of comonomer isomers of divinylbenzene (DVB) on the optical properties of inverse vulcanization products, revealing a promising avenue. By opting for DVB isomers over DIB, bulk polymers (> 1 mm thickness) showcased noteworthy LWIR transmission capabilities, approaching approximately 10% transparency¹⁷”

3. Reviewer: Thiol-ene and Thiol-yne reactions are known to proceed via thermal initiation (as well as other methods such as UV initiation). For references, see doi.org/10.1002/macp.200900442 and doi.org/10.1021/acsomega.8b00319. Therefore, the fact that the authors see an increase in the storage modulus as the temperature increases suggests that the polymer is not fully cured under UV light (as the authors point out). Did the authors consider doing both a UV post-cure of the polymers (ideally under inert atmosphere) as well as a comparative thermal post-cure? This question arises because one route might lead to more curing than the other, thus leading to a more stable product.

Response: We thank the comment to encourage us to better study our thiol-ene system. To answer the reviewer's question, we have conducted a kinetic study to show the thiol (alkene) conversion versus UV exposure time. Besides, we also did a post-curing using Formlabs curing machine to see how high the conversion can reach by UV curing and how it compared with the thermal post-curing results. Notably, we found out that tetraSH has a different curing behavior compared to polySH as the UV post-curing cannot help improve the conversion (~80%) for tetraSH but can increase the conversion of polySH system to around 90%. Such difference is mainly caused by the structure difference since tetraSH contains a number of secondary thiols but polySH only has primary thiols. In summary, we found that post-UV curing can help improve the monomer conversion for polySH but there is still space that can be improved as suggested by results of both DMA and DSC (see Fig. S11 and S12). On the other hand, the thermal post-curing can push the conversion over 90% for tetraSH and over 98% for polySH. That is, we think thermal post-curing is more favored in this study considering the final conversion.

One thing we would like to point out is that we do not have easy access to an instrument to run real-time FTIR experiments of samples cured under UV exposure. We used the same batch of material to prepare several samples with ~1 mm thick and then expose those samples under UV for different times. We tried our best to control all the conditions the same including the distance between the UV source and the sample and the UV power during exposure, but there will still be variations between each sample. As a result, the conversions fluctuated for samples cured under 2 min, 3 min, 4 min, etc., which should be more stable (or slightly increase) if the real-time

IR was used. However, it can still be concluded that those conversions are still kept in a stable range as shown in Fig S13 indicating the conversion tends to be constant after 1 min's UV curing.

In the revised manuscript, we added the content shown below:

“To enhance our comprehension of the curing kinetics and optimize thiol-ene conversion, we employed FTIR to monitor the consumption of both thiols and alkenes during UV exposure (Fig. S13). Our findings indicate that in both the tetraSH and polySH systems, the conversion of thiol rapidly stabilizes at approximately 70% (thiol) for tetraSH and 50% (thiol) for polySH within the initial 60 seconds of standard UV exposure (Fig. S13e & S13f). These conversions are maintained over the subsequent 9-minute period. In order to investigate the potential for post-UV curing to further enhance thiol-ene conversion, we conducted two additional post-UV curing steps. Initially, we extended the UV exposure by another 10 minutes while maintaining the same power. The tetraSH system exhibited consistent conversion rates for both thiol and alkene, whereas the polySH system demonstrated a slightly increased conversion (to ~55% for thiol). Subsequently, we subjected the samples to additional curing using a Formlabs post-curing machine (at room temperature for 12 hours). The tetraSH system's conversion remained unchanged, whereas the polySH system's thiol-ene conversion significantly increased to approximately 90% (thiol), highlighting distinctive UV curing kinetics for these two systems. This difference can be attributed to structural dissimilarities between tetraSH and polySH, as the latter contains only primary thiols while the former comprises both primary and secondary thiols.

During the initial 10-minute curing period, the tetraSH shows a higher reaction rate in comparison to the polySH system which might be due to the lower molecular weight and viscosity of tetraSH compared to polySH. However, the lower reactivity of secondary thiols in tetraSH compared to primary thiols becomes a limiting factor.⁴⁹ As the curing process progresses, the constrained mobility impedes the remaining secondary thiols from reacting effectively with alkenes under UV conditions. Conversely, in the polySH system, unreacted primary thiols continue to react with alkenes, albeit at a relatively slower rate. This continuous reaction capability enables the conversion to reach 87% through UV exposure. Nevertheless, while the DSC results for both the polySH-DVO2 and tetraSH-DVO2 systems display a nearly flat curve between 100 to 200 °C (Fig. S12c), DMA reveals a slight elevation in E' values for both systems at temperatures exceeding 160 °C (Fig. S11b). This suggests a potential for further improvement in thiol-ene conversion.”

“Besides UV curing, it is well-known that thiol-ene system can be cured by thermal curing with and without initiators.^{26,27,50}”

“Correspondingly, the thiol conversion for tetraSH-DVO2 and polySH-DVO2 were increased to 91% and 98%, respectively, as the thiol and alkene peaks became almost invisible in FTIR spectra (Fig. S13 a-d). Here, the thiol conversion of tetraSH-DVO2 also increased indicating that the increased temperature helps increase the molecule mobility which makes secondary thiol more easier to react with alkene.”

4. Reviewer: For the data in Figure 3, including a table with the specific values (e.g. Tg, stress/strain) would be very helpful.

Response: Thank you for this suggestion. We added a table (Table 1. Summary of thermal and mechanical properties of reported thiol-ene systems.) in the manuscript.

5. Reviewer: However, there are a number of published papers in which similar results have been obtained (though not using thiol-ene chemistry). That said, this reviewer would ask the authors to consider adjusting the language throughout the manuscript to better reflect this effort's place within the broader context of recent work detailing the synthesis of LWIR-transmissive polymers.

Response: Thanks for the comment. We went through the whole manuscript and adjusted the language to make sure that we do not overstate our work. For example: "It is important to acknowledge that the current multi-thiol-DVO system still possesses limitations concerning sample thickness in comparison to certain S-rich polymers generated through inverse vulcanization. However, the 3D printability inherent in this system confers upon it a distinctive value in the efficient fabrication of thin IR optics as and when required."

And also in the abstract:

"This UV-curable thiol-ene system provides a fast and convenient alternative for the fabrication of **thin** IR transparent objects."

Besides, we have added a table to compare our work to some of the reported organic IR transparent materials including transparency and thickness. Please see Table S1 (Table S1. IR transparency comparison between some reported material and multi-thiol-DVO material in this work. Ge and PE are listed as references.)

REVIEWERS' COMMENTS

Reviewer #1 (Remarks to the Author):

All my questions were addressed by the authors. No additional comments.

Reviewer #2 (Remarks to the Author):

The authors tried to clarify the novelty of the revised manuscript in the response letter, and I suggest it be accepted.

Reviewer #3 (Remarks to the Author):

This reviewer believes that the authors have sufficiently addressed the comments from the reviewers, and now supports the publication of this manuscript in Nature Communications.